# Integrated urban water management by coupling iron salt production and application with biogas upgrading

Zhetai Hu [1], Lanqing Li [2,3], Xiaotong Cen [1], Min Zheng [1], Shihu Hu [1], Xiuheng Wang [2,3], Yarong Song[1], Kangning Xu [4] & Zhiguo Yuan [1,5] ✉

Integrated urban water management is a well-accepted concept for managing urban water. It requires efficient and integrated technological solutions that enable system-wide gains via a whole-of-system approach. Here, we create a solid link between the manufacturing of an iron salt, its application in an urban water system, and high-quality bioenergy recovery from wastewater. An iron-oxidising electrochemical cell is used to remove $CO_2$ (also $H_2S$ and $NH_3$) from biogas, thus achieving biogas upgrading, and simultaneously producing $FeCO_3$. The subsequent dose of the electrochemically produced $FeCO_3$ to wastewater and sludge removes sulfide and phosphate, and enhances sludge settleability and dewaterability, with comparable or superior performance compared to the imported and hazardous iron salts it substitutes ($FeCl_2$, and $FeCl_3$). The process enables water utilities to establish a self-reliant and more secure supply chain to meet its demand for iron salts, at lower economic and environmental costs, and simultaneously achieve recovery of high-quality bioenergy.

Iron salts in various forms ($FeCl_2$, $FeCl_3$, $FeSO_4$ and $Fe_2(SO_4)_3$) are widely used in urban water management for a variety of purposes[1–3]. Most drinking water treatment plants rely on the use of iron- (or aluminum-) based coagulants for the removal of turbidity and natural organic matter[4,5]. Similarly, the addition of iron salts to sewer networks is widely applied to combat hydrogen sulfide ($H_2S$) induced sewer corrosion and odor[6–8], a notorious and multi-billion-dollar problem in sewer management[9,10]. Further, many wastewater treatment plants (WWTPs) rely on the dosing of iron- (or aluminum-) based salts for the removal of phosphate[11,12], and for improving sludge settleability and dewaterability[13,14]. Lastly, iron salts are also dosed to anaerobic digesters for reducing $H_2S$ in biogas[15,16]. These broad applications of iron salts lead to their consumption in large quantities. Indeed, iron salts represent a significant fraction of coagulants and flocculants consumed by the water industry, which had a global market of USD 6.4B in 2018 and is expected to reach USD 8.5B in 2023[17].

Iron salts currently used by the water industry are manufactured as a by-product of metallurgical processes. For example, the iron salts supplied in Australia are produced in the steel pickling process. Hydrochloric acid (HCl) or sulfuric acid ($H_2SO_4$) is used to remove iron oxides at the surface of steel, resulting in an acidic spent pickling liquor containing $FeCl_2$ or $FeSO_4$. These ferrous salts can be further converted to ferric salts, if needed, via the addition of a strong oxidant such as chlorine ($Cl_2$) or peroxide ($H_2O_2$)[18]. In some other parts of the world, $FeCl_2$ or $FeSO_4$ are produced from titanium ores containing iron, as a by-product in titanium dioxide ($TiO_2$) production, again involving the use of $Cl_2$/HCl or $H_2SO_4$[19]. Iron salts are also produced from iron ore, or by dissolving iron using HCl or $H_2SO_4$[20].

[1]Australian Centre for Water and Environmental Biotechnology, The University of Queensland, St Lucia, QLD 4072, Australia. [2]State Key Laboratory of Urban Water Resource and Environment, Harbin Institute of Technology, Harbin 150090, PR China. [3]School of Environment, Harbin Institute of Technology, Harbin 150090, PR China. [4]Beijing Key Laboratory for Source Control Technology of Water Pollution, Collage of Environmental Science and Engineering, Beijing Forestry University, Beijing 100083, China. [5]School of Energy and Environment, City University of Hong Kong, Hong Kong SAR, China. ✉e-mail: zhigyuan@cityu.edu.hk

With the current supply chain, the sources of iron salts are in most cases a long way away from where they are required, resulting in the need for long-distance transport. This increases the costs and environmental footprint of the chemical supply and poses significant occupational health and safety (OH&S) challenges due to the hazardous and corrosive nature of these chemicals. The current supply chain is susceptible to many factors, e.g., both UK and Germany water utilities are currently in shortage of iron salts due to supply chain interruptions, which had forced the local authorities to allow the discharge of partially treated sewage to the environments. It is of strategic importance for the water industry to establish local and environmentally friendly iron salt supplies that have higher supply chain security.

There is currently an on-going paradigm shift in wastewater management from pollutant removal to resource recovery. The recovery of bioenergy, in the form of biogas, is now widely implemented. Biogas produced at a WWTP is currently almost exclusively used locally for thermal and electrical energy generation[21–24]. However, limited by the relatively low electricity price and low value of thermal energy at most places, the value of biogas thus derived is generally low, especially considering the significant capital and maintenance costs associated with the gas engines[25]. The high-value uses of biogas, for example as a transport fuel or for injection into the natural gas grids[26,27], require the removal of $CO_2$, which typically constitutes 30–50% of biogas[28]. Various physical and chemical processes have been developed and applied to efficiently remove $CO_2$ from biogas thus achieving biogas upgrading[28–31]. However, they are often energy-inefficient and most leave behind materials requiring disposal or regeneration, potentially causing secondary pollution[29]. For example, $CO_2$ absorption using amine solutions results in degraded solvent that are toxic to both humans and the environment[32].

In this work, we propose and demonstrate an electrochemical method for manufacturing iron salts, a solution that effectively addresses two challenges simultaneously. The proposed method is fundamentally different from the existing method of chemical iron salts production[13,33]. The proposed method facilitates the establishment of a local iron salts supply chain and simultaneously broadens the range of biogas applications. Specifically, an iron-oxidizing electrochemical process is introduced to remove $CO_2$ from biogas, thus upgrading biogas. Concomitantly, as a $CO_2$ sink, $FeCO_3$ is produced,

which can be introduced to an urban water system as a substitute of the currently used iron salts, with comparable or superior performance. Mass balance assessment shows that the amount of $FeCO_3$ produced at a WWTP via this pathway meets the demand for iron salts by the catchment collecting and transporting wastewater to the plant. The economic and life-cycle assessments show that the supply pathway proposed in this study is more cost effective and more environmentally friendly than the current supplies.

## Results

### Electrochemical $CO_2$ removal from biogas and $FeCO_3$ production

The $CO_2$ removal tests were conducted in an electrochemical cell modified from a glass bottle, with two iron plates as the electrodes (Supplementary Fig. 1). NaCl at 2 g/L, sparged with the feed gas for about 30 min to strip dissolved oxygen, was used as the electrolyte. The feed gas contained $CO_2$ at ~40%, $CH_4$ (or $N_2$ as a non-explosive surrogate of $CH_4$) at ~60%, and trace levels of $H_2S$ at ~900 ppmv and $NH_3$ at ~270 ppmv in some tests.

Each 6 h test comprised a 2 h preparatory phase followed by a 4 h experimental phase (Fig. 1a, Supplementary Figs. 3–6). A current was supplied in the preparatory phase to produce $Fe^{2+}$ at the anode ($Fe \rightarrow Fe^{2+} + 2e^-$) and $OH^-$ at the cathode ($2H_2O + 2e^- \rightarrow 2OH^- + H_2$). In the absence of gas feeding, dissolved inorganic carbon ($CO_2$, $HCO_3^-$, and $CO_3^{2-}$) as well as $CO_2$ in the reactor headspace, resulting from the initial gas sparging, were removed (Fig. 1a, Supplementary Figs. 3–6) via reactions $CO_2 + H_2O \rightarrow HCO_3^- + H^+ \rightarrow CO_3^{2-} + 2H^+$; $Fe^{2+} + CO_3^{2-} \rightarrow FeCO_3$; $2H^+ + 2OH^- \rightarrow 2H_2O$. The continued current supply following $CO_2$ depletion led to pH elevation to the pre-selected set-point (i.e. 7.5, 8.0, 8.5, or 9.0) due to the on-going production of hydroxide (Fig. 1a, Supplementary Figs. 3–6).

In the subsequent experimental phase, the feed gas was continuously fed to the cell. The continuous $CO_2$ removal resulted in an upgraded gas containing substantially lower level of $CO_2$ (e.g. 6.1 ± 0.1% in Fig. 1a). Concomitantly, $H_2$ produced in the cathodic reaction evolved into the headspace replacing $CO_2$ removed (Fig. 1a, b, Supplementary Figs. 3–6). The current was manually adjusted to keep the electrolyte pH at the set-point (8.5 in Fig. 1, Supplementary Figs. 5, 7, and at other pH levels in Supplementary Figs. 3, 4, and 6).

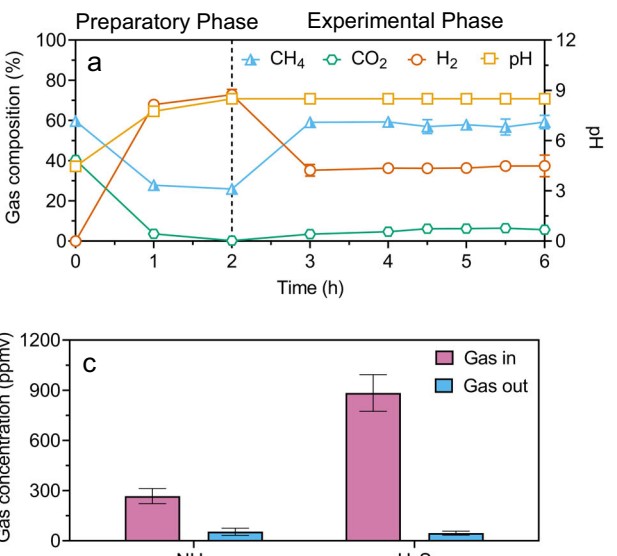
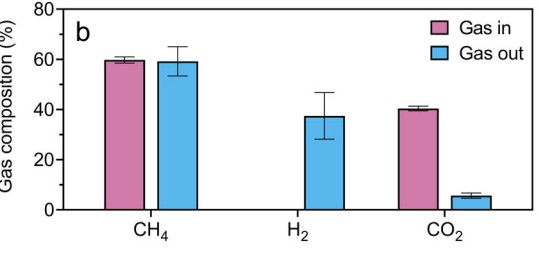
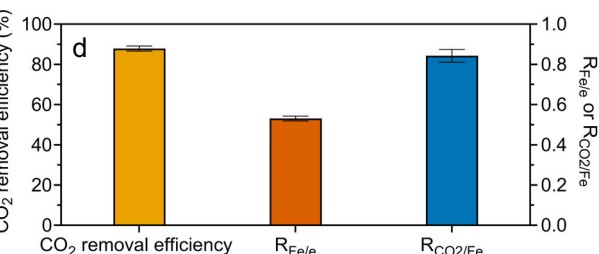

**Fig. 1 | Reactor performance in the tests at a pH set-point of 8.5. a** Contents of $CH_4$, $H_2$, and $CO_2$ in the headspace along with the electrolyte pH. **b** Average $CH_4$, $H_2$, and $CO_2$ concentrations in the feed and upgraded gas. **c** Average $NH_3$ and $H_2S$ concentrations in the feed and upgraded gas. **d** The Fe-to-electron ratio ($R_{Fe/e}$) and the $CO_2$-to-Fe ratio ($R_{CO_2/Fe}$). The vertical dotted line in (a) represents the start of continuous gas feeding (i.e. the commencement of experimental phase). All values are means ± standard in deviations of triplicate tests.

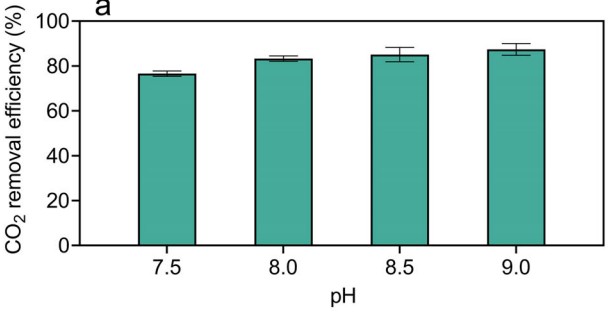

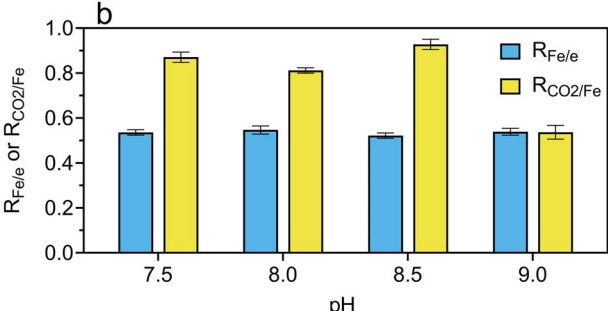

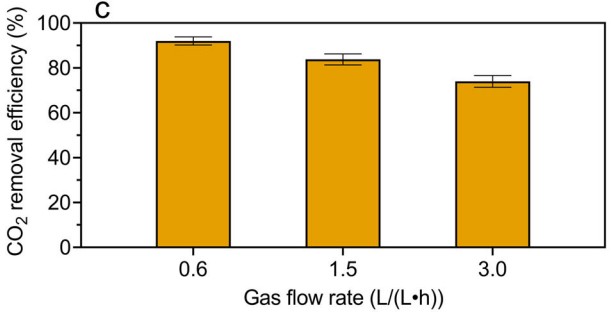

**Fig. 2 | Effects of pH and gas flow rate on cell performance. a** $CO_2$ removal efficiency in the tests at pH 7.5, 8.0, 8.5, and 9.0. **b** $R_{Fe/e}$ and $R_{CO_2/Fe}$ in the tests at pH 7.5, 8.0, 8.5, and 9.0. **c** $CO_2$ removal efficiency in the tests with gas flow rates of 0.6, 1.5, and 3.0 L/(L·h) at a pH set-point of 8.5. All values are means ± standard deviations of triplicate tests.

The concentrations of $NH_3$ and $H_2S$ in the gas were reduced from $267.5 \pm 26.1$ ppmv and $884.1 \pm 63.5$ ppmv to $54.3 \pm 12.3$ ppmv and $46.2 \pm 6.8$ ppmv, respectively (Fig. 1c), along with $CO_2$ removal. In contrast, $CH_4$ (or $N_2$) in the feed gas directly evolved into the headspace due to their low solubilities (Fig. 1a, b, Supplementary Figs. 2–7).

At pH 8.5, the ratio between the Fe oxidized and electrons transferred ($R_{Fe/e}$) was $0.52 \pm 0.01$ (mole/mole), suggesting the majority of electrons transferred were produced from Fe oxidation to $Fe^{2+}$. The ratio between $CO_2$ removed and Fe oxidized ($R_{CO_2/Fe}$) was $0.84 \pm 0.03$ (mole/mole), suggesting the majority of $Fe^{2+}$ produced was used for $CO_2$ removal (Fig. 1d). Overall, the results demonstrate the feasibility of $CO_2$, $H_2S$, and $NH_3$ removal from biogas using an iron-oxidizing electrochemical cell.

The cell performance is strongly pH dependent. Lower headspace $CO_2$ contents were achieved with the increase of pH (Fig. 2a, Supplementary Figs. 3–6). However, $R_{CO_2/Fe}$ decreased sharply when pH increased from 8.5 to 9.0 (Fig. 2b, Supplementary Table 1), indicating that a substantial fraction of ferrous ions produced was not combined with carbonate at pH 9.0, likely due to the formation of $Fe(OH)_2$ as an additional precipitate. Overall, pH 8.5 appears to be a favorable condition with relatively high $CO_2$ removal efficiency (i.e. $85.1 \pm 0.4\%$) and $R_{Fe/e}$ (i.e. $0.52 \pm 0.01$), and satisfactory $R_{CO_2/Fe}$ (i.e. $0.84 \pm 0.03$).

The gas flow rate also impacted the $CO_2$ removal efficiency (Fig. 2c). The $CO_2$ concentration in the upgraded gas increased with the gas flow rate from $3.2 \pm 0.3\%$ at 0.6 L/(L·h), to $6.2 \pm 0.1\%$ at 1.5 L/(L·h), and then to $9.5 \pm 0.5\%$ at 3.0 L/(L·h). An increase in the gas flow rate reduces the gas residence time[34], which decreases the $CO_2$ reaction time and reduces the $CO_2$ removal efficiency. The results suggest that a high level of $CO_2$ removal is possible by designing the reactor and the gas supply so that a satisfactory gas retention time and gas transfer rate is achieved.

The $FeCO_3$ produced, called E-$FeCO_3$ hereafter to be distinguished from the commercially available $FeCO_3$ (C-$FeCO_3$) that will later also be used in experimental studies, exists as solids in a slurry. The average particle size in the slurry produced in the cell is in the micron range with the $D_{10}$, $D_{50}$, and $D_{90}$ values being $6.9 \pm 0.6$,

$20.1 \pm 2.3$, and $46.7 \pm 3.2$ μm, respectively (Supplementary Fig. 8a, b). The particle size was measured as it likely influences the efficacy of E-$FeCO_3$ to react with sulfide or phosphate when added to wastewater or sludge, due to e.g. surface limitations or solids settling. Anaerobic storage of the slurry for up to 4 weeks did not significantly ($p = 0.73$) change the particle size distributions (Supplementary Fig. 8a, b). The particles, freshly produced or stored for up to 4 weeks, remained in suspension under turbulent conditions simulating those in sewers (Supplementary Fig. 8c). This means that the particles would remain suspended in sewage after in-sewer dosing, a desirable property for its in-sewer use.

Three crystalline iron species in the E-$FeCO_3$ slurry were identified to be siderite ($FeCO_3$), goethite (α-FeO(OH)), and hematite ($Fe_2O_3$) (Supplementary Fig. 9). Among these, $FeCO_3$ is the only compound containing $Fe^{2+}$, thus the measured fraction of $Fe^{2+}$ in total Fe ($86.2 \pm 3.9\%$) represents the fraction of $FeCO_3$ in all Fe-containing compounds. This is consistent with the measured ratio between $CO_2$ removed and Fe oxidized ($R_{CO_2/Fe}$), which is $0.84 \pm 0.03$.

## Application of E-$FeCO_3$ to wastewater and sludge management

The E-$FeCO_3$ slurry was added to anaerobic sewage, aerated activated sludge, and an anaerobic sludge digester to test its potential to remove sulfide and phosphate, despite $Fe^{2+}$ is in precipitates rather than as a dissolved ion. Dosed to anaerobic sewage, the E-$FeCO_3$ slurry quickly reduced the dissolved sulfide concentration in 0.5 h (Fig. 3a). The ratio between the sulfide removed and the Fe dosed, determined from the results in the underdosing tests, was $0.51 \pm 0.04$ g S/g Fe (Fig. 3a). Meanwhile, the wastewater pH was raised by 0.3 unit (Fig. 3d), caused by the release of carbonate from the E-$FeCO_3$ slurry. An increase in pH is favorable for sulfide and $Fe^{2+}$ precipitation[3,35]. Indeed, the dissolved sulfide concentration reduced to $0.08 \pm 0.02$ mg S/L when the E-$FeCO_3$ slurry was overdosed (Fig. 3a). The dosing of the E-$FeCO_3$ slurry to an anaerobic sludge digester controlled dissolved sulfide at $1.8 \pm 0.4$ mg S/L, compared to $30.5 \pm 1.9$ mg S/L in control (Fig. 3c), with $H_2S$ in biogas reduced from $1171.8 \pm 269.2$ ppmv (in control) to

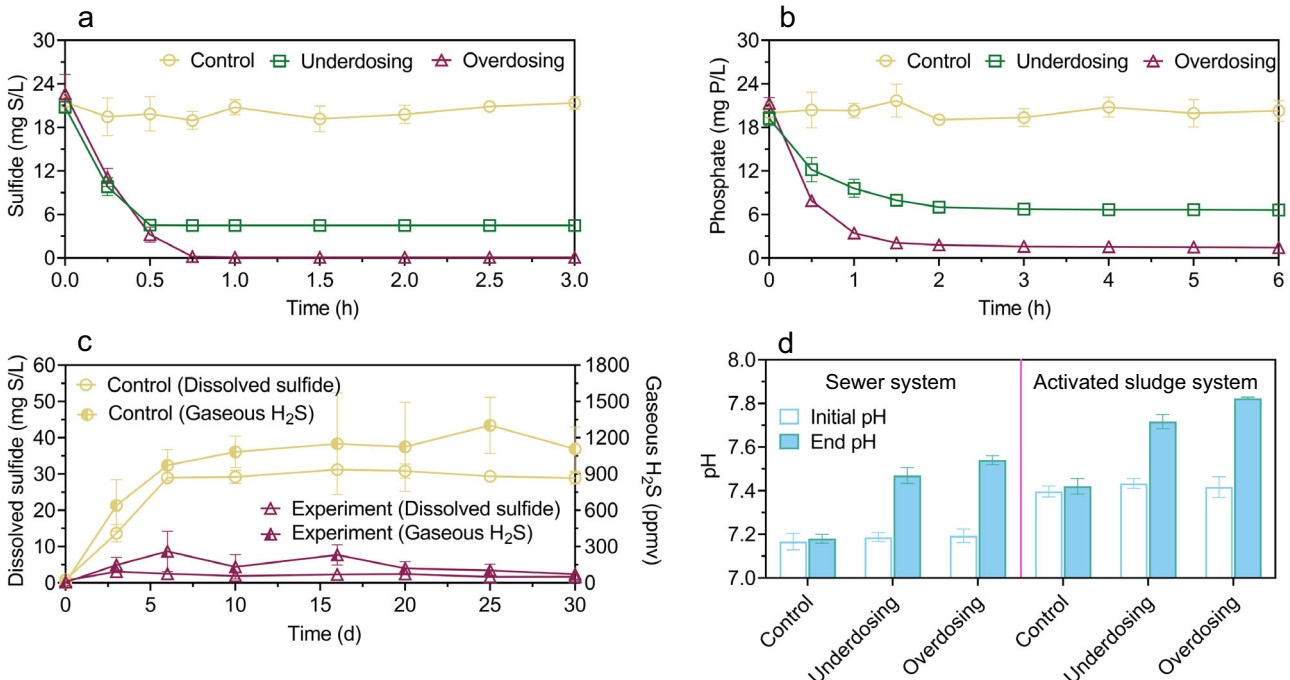

**Fig. 3 | Application of FeCO₃ slurry to wastewater, activated sludge, and anaerobic digester. a** Sulfide removal from sewage. **b** Phosphate removal in aerated activated sludge, **c** Sulfide removal in an anaerobic sludge digester, **d** pH at the end of each batch test in (**a**) and (**b**). The FeCO₃ slurry added was freshly produced at pH 8.5. All values are means ± standard deviations of triplicate tests.

85.7 ± 55.1 ppmv (in the experimental digester). Biogas production was not affected by E-FeCO₃ dosing (Supplementary Fig. 10).

Controlling effluent phosphate concentration at a low level is essential for impeding eutrophication. When added to aerated activated sludge, the E-FeCO₃ slurry significantly ($p < 0.01$) reduced the phosphate concentration within an hour (Fig. 3b). The ratio between phosphate removed and Fe dosed was $0.56 \pm 0.02$ g P/g Fe, determined from the results in the underdosing tests (Fig. 3b). A phosphate concentration of $1.62 \pm 0.09$ mg P/L was achieved in the overdosing tests (Fig. 3b). Meanwhile, the sludge pH in the experimental reactors was significantly ($p < 0.01$) higher than that in control (Fig. 3d).

The E-FeCO₃ slurry, after storage for 1, 2 and 4 weeks, showed similar sulfide removal performance to that of the fresh slurry, with the majority of sulfide removed within 1 h (to below 0.1 mg S/L). The sulfide to Fe ratios were $0.46 \pm 0.02$ g S/g Fe, $0.51 \pm 0.03$ g S/g Fe, and $0.48 \pm 0.02$ g S/g Fe, respectively (Supplementary Fig. 11). The sewage pH also significantly ($p < 0.01$) increased from ~7.2 to ~7.5.

Sewer networks, wastewater treatment processes, and anaerobic sludge digesters are interconnected in an urban wastewater system. The impacts of E-FeCO₃ dosing to sewer networks on the performance of downstream biological wastewater treatment and anaerobic sludge digestion processes were investigated via a series of batch experiments by adding E-FeCO₃-dosed sewage to aerated sludge, which was subsequently added to an anaerobic sludge digester. In-sewer E-FeCO₃ dosing resulted in phosphate removal in the aerated sludge at a ratio of $0.51 \pm 0.09$ mg P/mg Fe (Fig. 4b), following sulfide removal in the sewer reactor (Fig. 4a). This was likely due to the oxidation of FeS particles formed in the anaerobic sewer in the aerated activated sludge, as indicated by the increased sulfate concentration, resulting in a flowing-on effects of phosphate precipitation with the regenerated iron (Fig. 4b). Nitrification by the aerated sludge was not impacted by the wastewater amendments with the E-FeCO₃ slurry (Supplementary Fig. 12). The anaerobic digestion of the activated sludge receiving the E-FeCO₃-dosed sewage had negligible sulfide accumulation in the digester and biogas, despite complete sulfate reduction, in clear contrast to the control (Fig. 4c). These results suggest that the iron

originally dosed to the sewer reactor had a further flowing-on effects of sulfide control in the digester. The methane production performance was not impacted (Supplementary Fig. 13). The settleability and dewaterability of the sludge receiving E-FeCO₃ amended sewage were also found to be significantly ($p < 0.01$) improved by $36.9 \pm 2.7\%$ and $39.1 \pm 4.5\%$, respectively (Fig. 4d).

## Comparison of E-FeCO₃ with other iron salts in wastewater and sludge management performance

The experimental results provided compelling evidence for the efficacy of E-FeCO₃ in removing sulfide and phosphate, as well as its ability to improve sludge settleability and dewaterability. Further, the performance of E-FeCO₃ was compared with the C-FeCO₃, FeCl₂, and FeCl₃. C-FeCO₃ displayed negligible ability to remove sulfide or phosphate from wastewater/sludge, and limited ability to improve sludge settleability and dewaterability (Fig. 5). This could be attributed to its more stable crystalized structure in larger particles (Supplementary Fig. 14), which possibly reduced its reaction rate with sulfide and phosphate ions in the wastewater and sludge.

In contrast, E-FeCO₃, FeCl₂, and FeCl₃ were all effective in eliminating sulfide and phosphate and in enhancing sludge settleability and dewaterability (Fig. 5). Specifically, in anaerobic sewage, the dissolved sulfide control efficiencies of E-FeCO₃, FeCl₂, and FeCl₃ were $0.53 \pm 0.02$ g S/g Fe, $0.51 \pm 0.02$ g S/g Fe, and $0.56 \pm 0.03$ g S/g Fe, respectively (Fig. 5a). Dissolved sulfide concentrations below 0.1 mg S/L were achieved in the overdosing tests with all these iron salts (Supplementary Fig. 15a). Similarly, in the anaerobic sludge digesters, all three iron salts reduced the dissolved sulfide and gaseous H₂S concentrations to below 2 mg S/L and 200 ppmv, respectively, with nearly 90% reduction. (Fig. 5f). The methane production and sulfate reduction processes were not impacted (Supplementary Fig. 15e, f). E-FeCO₃, FeCl₂, and FeCl₃ also displayed similar efficiencies in phosphate removal from aerated sludge, at $0.47 \pm 0.02$ g S/g Fe, $0.51 \pm 0.02$ g S/g Fe, and $0.52 \pm 0.02$ g S/g Fe, respectively (Fig. 5c). Furthermore, the uses of E-FeCO₃, FeCl₂, and FeCl₃ increased the sludge settleability by $36.9 \pm 6.2\%$, $37.4 \pm 3.4\%$, and $49.7 \pm 3.6\%$,

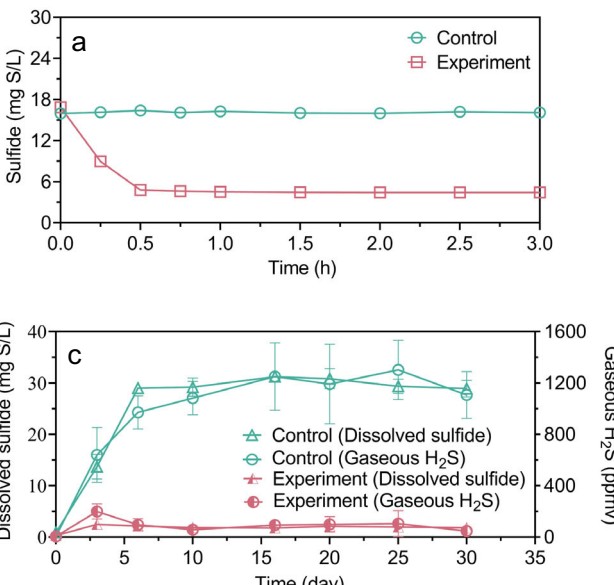

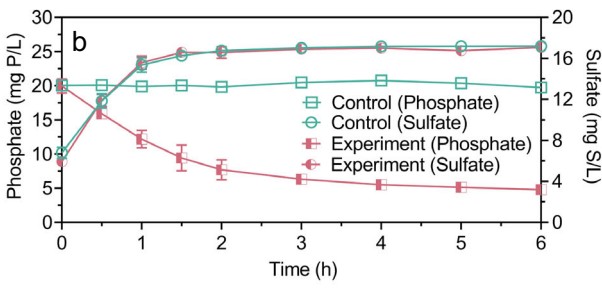

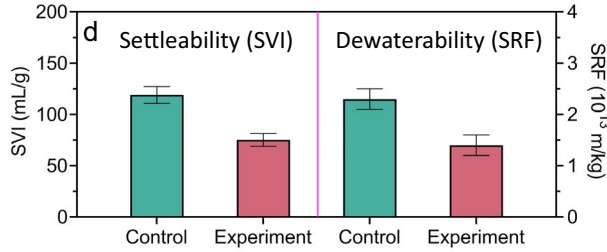

**Fig. 4 | Flowing-on effects of in-sewer dosing of E-FeCO₃ slurry on downstream wastewater and sludge treatment processes. a** Sulfide control in sewer. **b** Phosphate removal in aerated activated sludge. **c** Sulfide control in an anaerobic sludge digester. **d** Improvement to sludge settleability (sludge volume index, SVI) and dewaterability (specific resistance to filtration, SRF). All values are means ± standard deviations of triplicate tests.

respectively, and enhanced the sludge dewaterability by $55.0 \pm 1.5\%$, $57.9 \pm 5.3\%$, and $63.8 \pm 3.2\%$, respectively (Fig. 5h). Overall, the performance of E-FeCO₃ and FeCl₂ in sulfide and phosphate removal and in improving sludge settleability and dewaterability is similar, which is also similar to, or slightly lower than that of FeCl₃. $Fe^{3+}$ is able to oxidize sulfide (in addition to $Fe^{2+}$ and sulfide precipitation) and is also known to have stronger flocculating or coagulating capabilities compared to $Fe^{2+}$, which may explain the performance difference observed. The performance of FeCl₂ and FeCl₃ observed in these tests is comparable to that reported in literature (Supplementary Table 2), supporting the reliability of the results reported herein.

In contrast, the doses of E-FeCO₃, FeCl₂, and FeCl₃ induced different pH shifts. Dosages of ~30 mg Fe/L of E-FeCO₃, FeCl₂ and FeCl₃ to anaerobic sewage altered sewage pH by 0.15, −0.02, and −0.13 units, respectively (Fig. 5b and Supplementary Fig. 15b). A higher dose at ~90 mg Fe/L caused pH variations of 0.31, −0.49, and −0.66 units, respectively (Fig. 5b and Supplementary Fig. 15b). Similar pH variation patterns were also observed in the experiments with activated sludge (Fig. 5d and Supplementary Fig. 15d) and anaerobic sludge digesters (Fig. 5g). In all these cases, the provision of additional alkalinity via E-FeCO₃ addition, in comparison to the consumption of alkalinity via FeCl₂ or FeCl₃ dosage, is favorable, as will be further discussed later (Discussion Section).

In conclusion, E-FeCO₃ proves to be a suitable replacement for FeCl₂ and FeCl₃ for wastewater or wastewater sludge management, while C-FeCO₃ shows limited effectiveness.

## An integrated urban water management strategy

The experimental findings support an integrated urban water management strategy, comprising the production of E-FeCO₃ at a WWTP via biogas upgrading, and the dosing of E-FeCO₃ to the upstream sewer catchment for corrosion and odor mitigation with various beneficial flowing-on effects, and/or to various units in the WWTP to achieve phosphorous removal from wastewater, sulfide removal in the anaerobic sludge digester, and to improve sludge settleability and dewaterability (Fig. 6).

The wastewater biodegradable chemical oxygen demand (bCOD) concentration affects the amount of biogas produced, which subsequently determines the amount of E-FeCO₃ that can be produced. Mass

balance analysis shows that the amount of E-FeCO₃ that can be produced via biogas upgrading can meet the demand for iron salts for these purposes in the same catchment (Supplementary Table 3). Even for sewage with a moderate bCOD concentration of 300 mg/L and a moderate wastewater bCOD to methane conversion ratio of 7%, the iron salts produced would add 12 mg Fe/L of sewage (Supplementary Table 3), adequate for all the above-mentioned purposes[10,14].

A full economic analysis of the proposed process is not possible before the process is scaled up. An input-output analysis is performed for a hypothetical catchment and WWTP with a sewage flow rate of 120 ML/d (Supplementary Table 2). The plant produces biogas at 1641 m³/d assuming the influent contains 300 mg/L of bCOD, the upgrading of which produces E-FeCO₃ at 1470 kg Fe/d and upgraded biogas at 1,641 m³/d. Replacing FeCl₂ for sewer dosing and gasoline as a car fuel, respectively, these products entail a combined output value of A\$2.1 m/y. In comparison, the combined costs for the input materials (biogas, electricity, NaCl, and recycled iron) are estimated to be A\$0.64 m/y.

The life-cycle environmental impacts of the proposed process (Scenario A), with E-FeCO₃ replacing FeCl₂ for in-sewer dosing and the upgraded biogas as a car fuel, were compared with the status quo (Scenario B), with FeCl₂ produced from steel pickling and biogas used for combined heat and electricity production (Supplementary Fig. 16). Scenario A is further divided into A1 and A2 with the electricity for E-FeCO₃ production generated from biogas (A1) and from the current mix of primary energy sources in Australia (A2), respectively. Scenario B is also divided into B1 and B2 with FeCl₂ transported for 1000 km and 4000 km, respectively.

The status quo Scenarios B1 and B2 have negative environmental impacts against almost all indicators (Fig. 7), as the environmental impacts of FeCl₂ production and transportation could not be completely offset by the combined heat and electricity production from biogas, with indicators of Freshwater Eutrophication and Marine Eutrophication being two exceptions (Supplementary Fig. 17).

In contrast, Scenario A1 delivers positive or negligible environmental impacts in all categories (Fig. 7), owing to (1) the positive environmental impacts achieved with the replacement of gasoline with the upgraded biogas as a car fuel, and (2) the negligible or even positive (via CO₂ fixation) environmental impacts of E-FeCO₃ production.

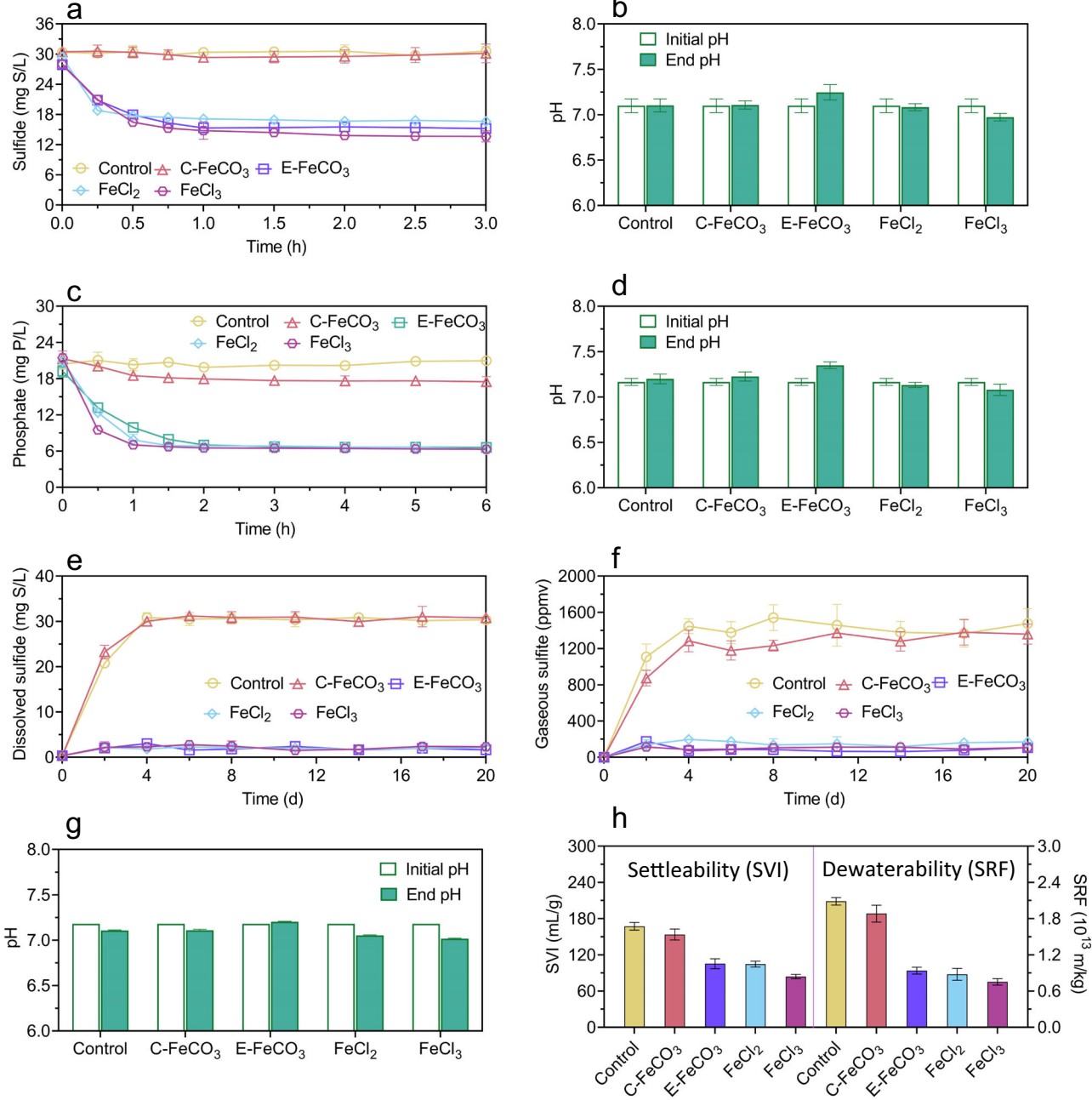

**Fig. 5 | Comparison of E-FeCO₃ with other iron salts in wastewater and sludge management performance. a** Sulfide control in sewer (underdosing). **b** pH at the beginning and end of each test in sewer (underdosing). **c** Phosphate removal in aerated activated sludge (underdosing). **d** pH at the beginning and end of each test in aerated activated sludge (underdosing). **e** Dissolved sulfide control in an anaerobic sludge digester. **f** Gaseous hydrogen sulfide (H₂S) control in an anaerobic sludge digester. **g** pH at the beginning and end of each test in an anaerobic sludge digester. **h** Improvement to sludge settleability (sludge volume index, SVI) and dewaterability (specific resistance to filtration, SRF). All values are means ± standard deviations of triplicate tests.

Consequently, A1 substantially outperforms B1 and B2 in most categories.

Different from A1, grid electricity is used to drive the electrochemical cell in A2. The current energy mix for power production in Australia has coal as a key component (54.9%). The impacts of coal use against several indicators could not be completely offset by the substitution of gasoline with upgraded biogas (Supplementary Fig. 17), due to the much lower impacts of gasoline on these indicators than coal. Nevertheless, A2 outperforms B1 and B2 against 13 of the 18 indicators. With the continued shift towards renewables in the energy mix, the environmental performance of A2 is expected to further improve in the years to come.

## Discussion

We are entering an era of circular economy. This requires us to improve our traditional approaches to cater to the requirement of sustainable development. This study showcases an integrated technological solution within an urban water system. It establishes a solid connection between biogas upgrading and the enhancement of wastewater and sludge management. Specifically, it offers the potential to protect sewer infrastructure, facilitate the removal and recovery of nutrients from wastewater, and reduce costs associated with sludge disposal. The experimental findings demonstrated the feasibility of this out-of-the-box solution, highlighting its ability to address multiple challenges simultaneously.

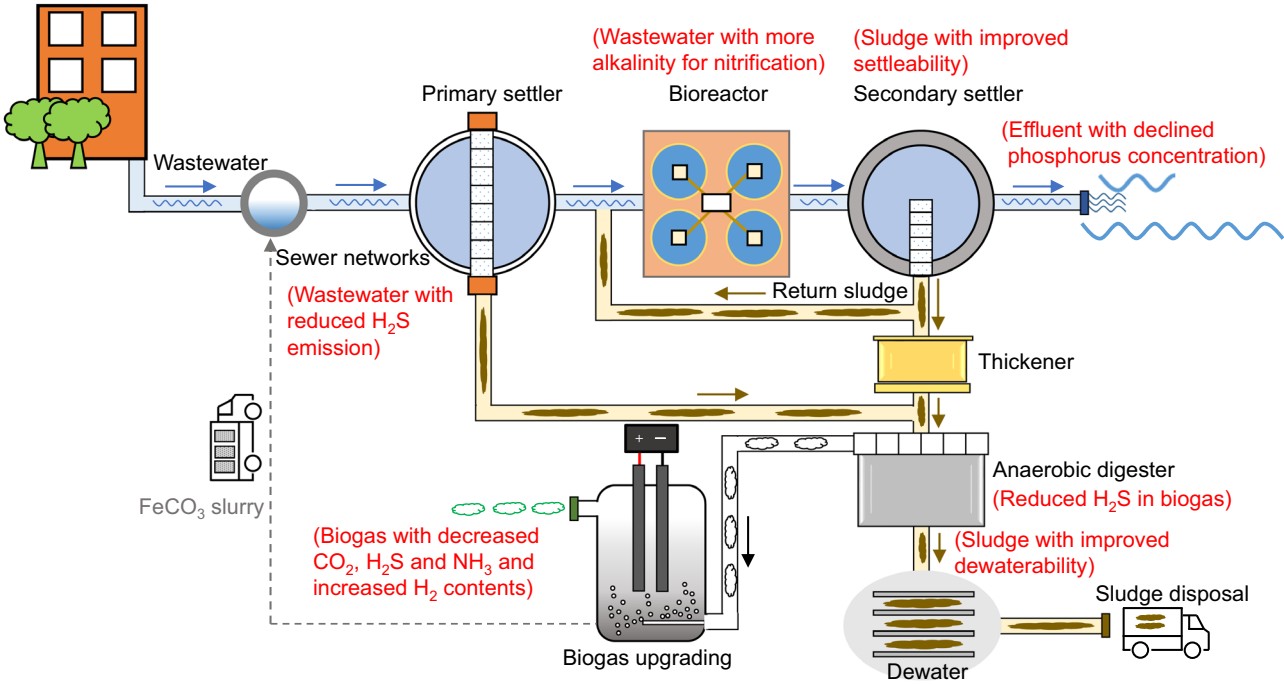

**Fig. 6 | Schematic drawing of our urban wastewater management system.** The system includes biogas upgrading and E-FeCO₃ production and application.

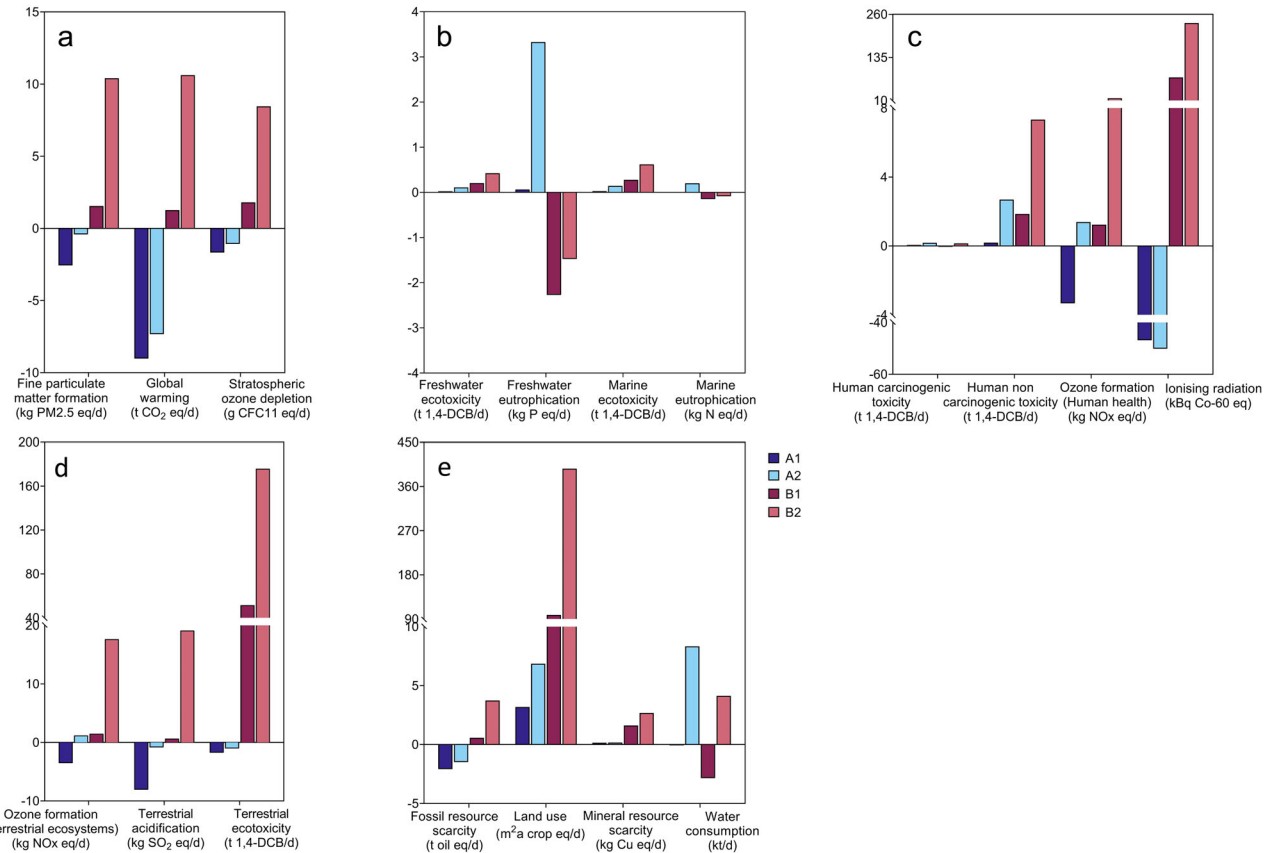

**Fig. 7 | Life cycle assessments of four different iron supply scenarios. a** Atmospheric environment impacts. **b** Aqueous environment impacts. **c** Human health impacts. **d** Terrestrial environment impacts. **e** Resource usage impacts.

An electrochemical cell was implemented to achieve the multiple goals. The construction of this electrochemical cell is straightforward, utilizing iron plates as electrodes and a NaCl solution as the electrolyte, without requiring complex materials or membranes. The iron should ideally be sourced from locally recycled iron. If this is not available, it could be imported from iron manufactures. Even in the latter case, the transport costs would be greatly reduced. As an example, the $FeCl_2$ supplied in Australia is a solution containing 12% iron. Theoretically, transporting iron as iron plates instead of this iron salt solution would substantially reduce the transportation cost, with a ~88% reduction in weight and a ~98% reduction in volume. In addition, this electrochemical cell can be integrated into existing wastewater treatment systems, following the AD process (Fig. 6). Furthermore, the system is easy to operate, and the control logic is simple. By adjusting the current to keep the electrolyte pH at a pre-selected level (recommended to be around 8.5), the amount of $OH^-$ produced is ensured to just meet the demand for $CO_2$ conversion to $CO_3^{2-}$ and its subsequent removal as $FeCO_3$.

Serving as scarifying electrodes, the Fe plates need to be replaced regularly, with an interval determined by the reactor size, the plate spacing, and the $CO_2$ loading rate. In the example used in the input-output analysis, the $CO_2$ loading rate is estimated to be in the range of 656−1750 $m^3$/d, given a bCOD concentration range of 300−800 mg/L (Supplementary Table 4). Assuming a total electrochemical cell volume of 200 $m^3$ and an iron plate spacing of 1 cm, the Fe plate replacement interval is estimated to be 4−10 months (Supplementary Table 4), which is reasonable.

The overall process comprises three key steps, namely the electrochemical production of $Fe^{2+}$ and $OH^-$, dissolution of $CO_2$ and its subsequent conversion to $CO_3^{2-}$ under alkaline conditions, and the precipitation of $Fe^{2+}$ and $CO_3^{2-}$ as $FeCO_3$. Among these, the $CO_2$ transfer from the gas bubbles to the electrolyte is the rate-limiting step, which determines the rate of the overall electrochemical system. The $CO_2$ mass transfer rate is influenced by the reactor configuration, the gas flow rate, the gas bubble size, and the operating pH. In our proof-of-concept experiments at an optimal pH of 8.5, ~85% of $CO_2$ in the feed gas was retained in the reactor as $FeCO_3$, while the remaining ~15% remained in the upgraded gas. The $CO_2$ removal efficiency can be further improved through reactor engineering and gas flow rate control (Fig. 2c). $CO_2$ removed from the feed gas was replaced by $H_2$ generated at the cathode with a molar ratio of 1:1. The energy content of $H_2$ in the upgraded biogas partially recovers the electricity energy invested.

This study proposed and experimentally demonstrated, that E-$FeCO_3$, despite in a solid form as particles, can replace soluble iron salts for wastewater management. However, the industry survey conducted in Australia and a more recent comprehensive literature review both showed the C-$FeCO_3$ had not been applied to wastewater and sludge management[7,36]. Our comparative experiments showed that the C-$FeCO_3$ is ineffective in sulfide or phosphate precipitation (Fig. 5), due to its stable crystalised structure in larger particles (Supplementary Fig. 14). The low reactivity of C-$FeCO_3$ limits its applications in urban water management.

Compared to $FeCl_2$ and $FeCl_3$, the dosage of E-$FeCO_3$ to wastewater or sludge causes a slight rise rather than a drop of pH, as in the case of $FeCl_2$ and $FeCl_3$ dosing. This is because the dosage of E-$FeCO_3$ provides additional alkalinity. In comparison, the dosage of $FeCl_2$ and $FeCl_3$ consumes alkalinity. In sewer networks, an increase of sewage pH is desirable as it shifts the $H_2S$ and $HS^-$ equilibrium towards $HS^-$ ($H_2S \leftrightarrow HS^- + H^+$), and hence reduces the transfer of $H_2S$ from the liquid to the gas phase. In fact, alkali in the form of e.g. $Mg(OH)_2$ is often added to sewage in areas where $H_2S$ is problematic[7,36], clearly illustrating the importance of pH elevation for $H_2S$ control in sewers. In the wastewater treatment process, the additional alkalinity provided via E-$FeCO_3$ dosage is potentially beneficial for nitrification,

particularly for wastewaters containing relatively low levels of alkalinity. Although denitrification partially regenerates alkalinity consumed by nitrification, wastewater in some parts of the world still does not contain alkalinity at a level enabling satisfactory nitrogen removal[37,38]. In such cases, E-$FeCO_3$ should be a better source of iron than $FeCl_2$ or $FeCl_3$. Also, pH stability is critical for anaerobic sludge digesters. The additional alkalinity provided with E-$FeCO_3$ dosage helps improve pH stability in the digesters.

$Fe^{3+}$ salts are sometimes dosed to the primary settling tank in a wastewater treatment plant, or to the secondary effluent to remove phosphate. These cannot be replaced with $Fe^{2+}$-based salts including E-$FeCO_3$. However, E-$FeCO_3$ (indeed any other $Fe^{2+}$-based salts as well) could be added to the aeration basin to remove phosphate (Fig. 2).

The majority of iron dosed to the wastewater or the wastewater sludge will end in the biosolids as solid iron salts including iron-phosphate compounds, e.g. $Fe_3(PO_4)_2$ (vivianite)[39], which increases sludge production. However, as demonstrated in this as well as previous work, the addition of iron salts helps improve sludge settleability and dewaterability, which means additional sludge production may not necessarily lead to increased sludge disposal costs.

Numerous engineering aspects require further research in the upscaling of the process. In this proof-of-concept study, biogas was distributed to a short reactor using a 0.5 mm diameter needle. In full-scale applications, we expect that the biogas will be provided via a gas distribution system generating microbubbles in a relatively tall reactor. The reactor should be designed such that an adequate gas retention time is achieved for $CO_2$ in the up-traveling gas bubbles to diffuse into the liquid phase and dissociate as bicarbonate and carbonate, before the gas bubbles reach the reactor headspace releasing $CH_4$ and $H_2$. The electrode should be designed such that $Fe^{2+}$ can be produced at a rate required for $CO_2$ removal with an acceptable voltage and power consumption. Also, experiments should be performed at full-scale sewer networks, wastewater treatment plants, and anaerobic sludge digesters, operated in a variety of conditions, to test the effectiveness of the E-$FeCO_3$ slurry for the intended purposes.

## Methods

Numerous experiments were conducted to demonstrate the proposed concepts. The overall structure of the experimental design is shown in Supplementary Fig. 18, with details of each experiment described below.

### Electrochemical cell setup and operation

The $CO_2$ removal tests were conducted in a modified glass bottle with a total volume of 325 mL in a fume hood in a temperature-controlled ($22 \pm 1\,°C$) laboratory (Supplementary Fig. 1). The reactor was sealed to ensure gas-tightness and mixed with a magnetic stirrer at a speed of 300 rpm. Two iron plates (mild steel, Harding Steel), served as anode and cathode, respectively, were placed in parallel and fixed to the lid of the bottle, with an interelectrode gap of 1.0 cm. The dimensions of the iron plates were 15 cm × 1.4 cm × 0.3 cm. Each iron plate was submerged at a depth of 3.5 cm in the electrolyte, achieving a submerged surface area of 11.9 $cm^2$. Iron oxidation was achieved by controlling the electrochemical cell current via a bench power supply (72-2685, TENMA, China). The feed gas was diffused into the electrolyte via a 0.5 mm diameter needle. Due to safety concerns, the feeding gas in all but one test comprised ~60% $N_2$ and ~40% $CO_2$ with $N_2$ as a proxy of $CH_4$ as both have a low solubility. In one test, the feed gas comprised ~60% $CH_4$ and ~40% $CO_2$, for comparison with results from tests with $N_2$, as well as trace levels of $H_2S$ ($884.1 \pm 63.5$ ppmv) and $NH_3$ ($267.5 \pm 26.1$ ppmv) to evaluate the capability of the electrochemical cell to remove these contaminants. The feeding gas flow was controlled with a gas flow controller (Bronkhorst, Netherlands), with the 'upgraded gas' collected with a 5 L gas bag connected to the reactor outlet. In each test, 200 mL of 2 g/L of NaCl solution, prepared using

tap water, was used as the electrolyte after being sparged with the feed gas for 30 min at a flow rate of 0.1 L/min, to remove the residual dissolved oxygen (DO). pH in the reactor was monitored with a portable pH meter (miniCHEM, Labtek). The reactor has sampling ports for gas, liquid, and solids sampling, as shown in Supplementary Fig. 1.

## Electrochemical $CO_2$ removal and E-$FeCO_3$ production

The $CO_2$ removal efficiency of the cell was evaluated in triplicate at pH 7.5, 8.0, 8.5 and 9.0, respectively, via a series of batch tests. The tests at pH 8.5 were repeated with gas feed composition changed from $N_2$ (~60%) and $CO_2$ (~40%) to $CH_4$ (~60%), $CO_2$ (~40%), $H_2S$ (884.1 ± 63.5 ppmv) and $NH_3$ (267.5 ± 26.1 ppmv). Each test lasted for 6 h, comprising 2 h of preparatory phase, and 4 h of experimental phase. Initially, 200 mL of oxygen-free electrolyte was added into the reactor, leaving 125 mL as the headspace. In the preparatory phase, a current was supplied to the cell in the absence of a gas supply. pH in the reactor was progressively elevated to the pre-specified level (i.e. 7.5, 8.0, 8.5 or 9.0) due to the on-going production of hydroxide (along with $H_2$) in the cell. The subsequent experimental phase commences when the pH set-point was reached, during which the feed gas was fed into the reactor at a rate of 5 mL/min. The current in the experimental phase was further manually adjusted so that the pH was maintained at the set-point (i.e. 7.5, 8.0, 8.5 or 9.0). This adjustment was only needed at the beginning of the phase, as pH remained stable once a suitable current was found, due to the following reactions: $Fe + 2H_2O \rightarrow Fe^{2+} + 2OH^- + H_2$ and $CO_2 + H_2O + Fe^{2+} \rightarrow FeCO_3 + 2H^+$.

Gas samples were taken from the headspace of reactor with a 100 μL syringe hourly in the first 4 h, and then every half hour in the last 2 h. The liquid and solid samples were taken hourly for the analysis of iron concentration. The voltage was recorded every 5 min manually. About 50 mL Fe-containing slurry freshly produced at pH 8.5 was collected for XRD analysis, a further 3 mL sample was collected for particle size distribution analysis. Finally, at the end of each test, all the liquid and solid content in the reactor was transferred into a 200 mL oxygen-free sealed bottle for further experiments as described below.

One additional set of experiments was conducted at pH 8.5, aimed to evaluate the effect of gas flow rate on the cell performance. The test lasted for 9 h, comprising a 2 h preparatory phase with pH elevated to 8.5 in the absence of a gas supply, and a 7 h experimental phase, during which the gas flow rate was stepwise increased from 2 mL/min (3 h) to 5 mL/min (2 h), and further to 10 mL/min (2 h). The current was manually adjusted following each change of the gas flow to ensure a constant pH at 8.5. Gas samples were taken hourly in the first 4 h, and then every half hour in the following 5 h.

## E-$FeCO_3$ as an iron salt to support urban wastewater management

Two sets of experiments were designed to assess the suitability of E-$FeCO_3$ produced in biogas upgrading for supporting urban wastewater management. The first set was designed to assess the effects of E-$FeCO_3$ slurry dosing to sewers on sulfide control, to a biological wastewater treatment reactor on phosphate removal, and to an anaerobic sludge digester on sulfide control. In the second set, the flow-on effects of in-sewer dosed E-$FeCO_3$ slurry on the performances of biological wastewater treatment system and anaerobic digestion were investigated, noting that an urban wastewater system is an integrated system.

Wastewater was collected from a local domestic wastewater pump station (Brisbane, Australia), and stored at 4 °C prior to use to minimize changes in wastewater characteristics. It had a pH of 7.1–7.4 and contained total COD at 400–600 mg/L including soluble COD at 220–310 mg/L, phosphate at 4–7 mg P/L, iron at 0.1–0.3 mg Fe/L, sulfate at 10–20 mg S/L, sulfide at 5–10 mg S/L, and undetectable levels of oxygen. Activated sludge was collected from a local WWTP (Brisbane, Australia), with a mixed liquor suspended solids (MLSS) and a mixed

liquor volatile suspended solid (MLVSS) concentration of 13.2 ± 0.1 g/L and 10.6 ± 0.1 g/L, respectively. Anaerobically digested sludge was collected from a laboratory anaerobic digestion reactor, with the total solid (TS) and volatile solid (VS) concentrations of 20.6 ± 0.1 g/L and 16.3 ± 0.1 g/L, respectively.

The E-$FeCO_3$ slurry produced at pH 8.5 was used to conduct all these experiments.

**The effect of E-$FeCO_3$ slurry on sulfide control in sewer.** For each sulfide removal experiment in sewer, wastewater of 290 mL was filtered using disposable millipore filter units (0.45 μm), and then transferred into a 300 mL sealed bottle. The bottle was stripped with pure nitrogen gas for 30 min to further remove dissolved oxygen. A sulfide stock solution ($Na_2S \cdot 9H_2O$ of ~1.5 g S/L) of 5 mL was then added to the bottle to increase the sulfide concentration to ~25 mg S/L, followed by the addition of 1 M HCl to obtain a pH of 7.2, typical of domestic wastewater. After that, a pre-determined amount of the E-$FeCO_3$ slurry was added to each experiment to achieve a pre-designed initial iron concentration (described below). To guarantee there was no headspace during the experiment, two syringes filled with filtered and oxygen-free wastewater, were connected to the reactor to replenish the reactor after sampling. Each test lasted for 3 h, during which the reactor was mixed with a magnetic stirrer at 300 rpm. Liquid samples were taken before E-$FeCO_3$ dosing, and every 15 min in the first hour after the dosing, and then every 30 min, for the measurement of dissolved sulfide. pH in the reactor was monitored with a portable pH meter and recorded manually at the same intervals. An additional sample was taken at the end of each test to measure the total iron concentration.

Two different initial Fe levels, namely 30 and 90 mg Fe/L, were used in the above-described experiments. According to the theoretical reaction stoichiometry ($Fe^{2+} + S^{2-} \rightarrow FeS \downarrow$), an initial Fe concentration of 30 mg/L is insufficient for removing the sulfide initially present in the wastewater (~25 mg S/L), and hence the ratio between sulfide removed and Fe added could be determined. In contrast, Fe would be in excess for an initial Fe concentration of 90 mg Fe/L, and hence the lowest achievable sulfide concentration can be determined.

The above experiments were performed with both freshly produced E-$FeCO_3$ slurry i.e. with experiments undertaken within 1 day following the E-$FeCO_3$ production, and E-$FeCO_3$ slurry stored in a sealed serum bottle at a temperature-controlled (22 ± 1°C) laboratory for 1, 2 and 4 weeks to determine the impact of E-$FeCO_3$ storage on the sulfide removal performance.

**Suspension of E-$FeCO_3$ particles in sewer.** The electrochemically produced E-$FeCO_3$ was in a slurry. For its use in sewers for sulfide control, it should remain in suspension after addition to sewage under in-sewer hydrodynamic conditions. Batch tests were therefore conducted in a 200 mL reactor that was mixed with a magnetic stirrer at an intensity that creates turbulence, as described by the Reynolds number, similar to that in gravity or rising main sewers. At the start, 198 mL of tap water, stripped with nitrogen gas for 30 min to remove the DO, was transferred to the bottle, followed by the injection of 2 mL E-$FeCO_3$ slurry with a syringe. The iron concentration thus obtained is estimated to be ~100 mg Fe/L, simulating an overdosing situation. Each test lasted for 30 min. Liquid samples were taken through the middle sampling port, immediately after the E-$FeCO_3$ dosing and at the end of the test, for the measurement of total iron concentration. Identical iron concentrations would indicate the absence of E-$FeCO_3$ settling.

The experiments were performed with both freshly produced E-$FeCO_3$ slurry and E-$FeCO_3$ slurry stored for 1, 2, and 4 weeks to determine the impact of E-$FeCO_3$ storage on the sulfide removal performance. The particle size distributions in the stored slurries were measured prior to use.

**The effect of E-FeCO$_3$ slurry on phosphate removal during wastewater treatment.** For each phosphate removal test, activated sludge of 100 mL was mixed with 400 mL filtered wastewater, with the mixture transferred to a 1 L bottle. A phosphate stock solution (5 g P/L of KH$_2$PO$_4$) of 1.5 mL was then added to the bottle to increase the phosphate concentration to about 20 mg P/L. The E-FeCO$_3$ slurry (~10 g Fe/L) of about ~0.8 and ~3.5 mL was dosed to different experimental bottles to obtain two levels of initial Fe concentrations, namely ~16 and ~70 mg Fe/L. Control tests were also conducted without iron dosing. Each test lasted for 6 h, during which the DO concentration was controlled at 2.0–3.0 mg O$_2$/L with a programmable logic controller (PLC) via on/off control of the air flow. The reactor was mixed with a magnetic stirrer at 300 rpm. Liquid samples were taken before E-FeCO$_3$ dosing, and every 0.5 h in the initial 2 h, and then hourly, for the measurement of phosphate concentration. The reactor pH was monitored with a portable pH meter and recorded manually with the same intervals. An additional sample was also taken at the end of each test for the measurement of the total iron concentration.

**The effect of E-FeCO$_3$ slurry on sulfide control in anaerobic digestion.** The effect of E-FeCO$_3$ on sulfide control in anaerobic digestion was evaluated via biochemical methane potential (BMP) tests, conducted according to the standard procedure[40]. Specifically, about 20 mL thickened activated sludge (TS: 21.3 ± 0.1 g/L; VS: 17.7 ± 0.1 g/L) was mixed with ~40 mL inoculated digested sludge (TS: 20.6 ± 0.1 g/L; VS: 16.3 ± 0.1 g/L), and then transferred into a 100 mL sealed bottle. A blank test was also carried out using the feeding of ~20 mL wastewater and ~40 mL inoculated digested sludge. After that, the sealed bottle was stripped with pure nitrogen gas for 10 min to remove the residual oxygen. The E-FeCO$_3$ slurry (~10 g Fe/L) of about ~0.8 mL was dosed to the experimental bottles to obtain an initial Fe concentrations of ~80 mg Fe/L. After that, a sulfate stock solution (Na$_2$SO$_4$ of ~1.5 g S/L) of 1.0 mL was added into the reactor to increase the sulfate concentration to about 25 mg S/L, followed by the addition of 1 M HCl to adjust the reactor pH to ~7.5, typical for anaerobic sludge digester. Afterwards, all the BMP bottles were incubated in a temperature-controlled (37 ± 1 °C) incubator. The BMP tests lasted for about 30 days until almost no further increase of biogas was detected. Gas samples were taken every 2 days in the initial 10 days, and every 5 days to the end, for the measurement of the content of N$_2$, CH$_4$, and CO$_2$ in the biogas. The volume of biogas produced in each BMP bottle was also measured at the same intervals. The gas pressure in each BMP bottle was regularly assessed using a manometer (Testo, Australia) prior to each sampling event. The volume of newly generated biogas was determined by calculating the difference in gas pressure between two consecutive sampling events. Gas and liquid samples were taken every 5 days for the measurement of inorganic sulfur species. An additional sludge sample was also taken at the end of each test for measuring sludge dewaterability.

**The flow-on effects of in-sewer dosed E-FeCO$_3$ slurry on downstream wastewater and sludge treatment.** The effect of in-sewer dosed E-FeCO$_3$ slurry on the biological wastewater treatment process was investigated in two steps, namely sulfide removal in sewer followed by phosphate removal during aerobic treatment of the E-FeCO$_3$-receiving wastewater with activated sludge. The sulfide removal step was performed as per the previous description, with freshly produced E-FeCO$_3$ slurry. The initial sulfide and Fe concentrations in this test were ~18 mg S/L and ~20 mg Fe/L, respectively, ensuring that E-FeCO$_3$ was not in excess. After the 3 h sulfide removal test, the 300 mL E-FeCO$_3$-dosed sewage was fed to 300 mL activated sludge which was prepared by mixing 150 mL activated sludge with the raw wastewater at a ratio of 1:1 (v/v). A phosphate stock solution (5 g P/L of KH$_2$PO$_4$) of 3.0 mL was then added to the bottle to increase the phosphate concentration to about 25 mg P/L. Each test, with the mixed liquor aerated,

lasted for 6 h, with the operational and monitoring procedures identical to those applied in the above-described phosphate removal test.

The effect of in-sewer dosed E-FeCO$_3$ slurry on the anaerobic sludge digestion was investigated in three steps, sulfide removal in sewer, phosphate removal during wastewater treatment, and sulfide control in anaerobic digestion. The first two steps were similar to the above-described experiments investigating the flow-on effect on phosphate removal, with the following differences. The initial sulfide, Fe, and phosphate concentrations in this experiment were much higher than those in the above-described experiments, being about 200 mg S/L, 300 mg Fe/L, and 300 mg P/L, respectively. This is because the in-sewer dosed Fe would accumulate in the activated sludge in a practical scenario. It was reported that Fe can accumulate at a concentration 20× that in the wastewater[41]. Following the P-removal test, a sludge sample of 100 mL was harvested for the measurement of sludge settleability, with the remaining sludge centrifuged at 700 × g for 3 min. In the third step, the concentrated sludge (~15 g VS/L) was used as the feed for BMP tests, to evaluate the effect of in-sewer dosed E-FeCO$_3$ on sulfide control in anaerobic sludge digestion. The operational conditions were similar to that mentioned in the Section on The effect of FeCO$_3$ slurry on sulfide control in anaerobic digestion.

**Comparison of E-FeCO$_3$ with other iron salts in wastewater and sludge management performance.** The performance of E-FeCO$_3$, C-FeCO$_3$, FeCl$_2$, and FeCl$_3$ in sulfide and phosphate removal from wastewater/sludge, and in sludge settleability, and dewaterability enhancement were compared via parallel experiments. The four iron salts were separately dosed to anaerobic sewage, aerated activated sludge, and anaerobic sludge digester, respectively. The operational conditions and experimental procedure were as described in the first set of experiments in Section E-FeCO$_3$ as an iron salt to support urban wastewater management. The C-FeCO$_3$ utilized in this study was procured from Lianyungang Huaihua International Trade Co., LTD. Additionally, the FeCl$_2$·4H$_2$O and FeCl$_3$·6H$_2$O reagents were acquired from Westlab, Australia.

**Chemical analysis**
The detection methods used in study, including MLSS, MLVSS, TS, VS, sludge volume index (SVI), TCOD, SCOD, gaseous CH$_4$, CO$_2$, and H$_2$, total Fe, and specific resistance to filtration (SRF), have been elaborated in Supplementary Table 5. Liquid samples were taken using a syringe and filtered through disposable Millipore filter units (0.22 μm, Millipore, Millex GP) for the analyses of ammonium, nitrite, nitrate, phosphate and inorganic sulfur species (i.e. sulfide, sulfate, silfite and thiosulfate). Ammonium, nitrite, nitrate, and phosphate were analysed using a flow injection analyzer (Lachat Instrument, Milwaukee, Wisconsin), and the sulfur species were measured by Ion Chromatography with an ultraviolet (UV) and conductivity detector (Dionex ICS-2000)[42]. Particle size was measured using dynamic light scattering (Zetasizer Nano ZS, Malvern Instruments). SRF, a common index of sludge dewaterability, was analyzed by using a multi-couple measuring device, as described in literature[43]. The XRD patterns were generated using an X-ray diffractometer (Bruker D8). Prior to XRD measurement, the Fe-containing slurry was dried under vacuum conditions (−50°C, 0.1 mbar), and then ground into powder under anaerobic condition.

**Life cycle assessment (LCA)**
The life cycles of two different iron salt supply scenarios for a hypothetical 120 ML/d WWTP were evaluated in this study (Supplementary Fig. 14). Scenario A represents the E-FeCO$_3$ approach proposed in this study, including the use of upgraded biogas to replace gasoline as car fuel and the use of E-FeCO$_3$ to bring multiple benefits to the wastewater treatment system. Scenario B represents a status quo FeCl$_2$ supply approach, including the production and transportation of FeCl$_2$ as well as the utilization of biogas for combined power and heat

production. Further details of the scenario modeling are provided in Supplementary Table 6 and 7. The impact assessment was carried out using the ReCiPe 2016 Midpoint (H) method in the openLCA 1.10 software. In total, the software estimates 18 environmental impacts. To address the uncertainty of model parameters, 10,000 Monte Carlo simulations were conducted. The detailed uncertainty analysis results are shown in Supplementary Table 8.

## Statistical analysis

To identify the significant difference between experimental and control tests, a student $t$-test was performed in Microsoft Excel. If the $P$-vale is below 0.05, it means the difference is significant, and vice versa.

## Data availability

The authors declare that the data supporting the findings of this study are available within the paper and its supplementary information files. Source data are provided with this paper.

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

## Acknowledgements

We acknowledge X. Lu., for help with sludge dewaterability measurement and X. Huang, for assistance with XRD and SEM data assessment. Z.Y. was a recipient of the Australian Research Council Australian Laureate Fellowship (FL170100086) and is a Global STEM Scholar funded by the Government of the Hong Kong Special Administrative Region. Z.H. thanks the support from the China Scholarship Council (CSC). M.Z. acknowledges the joint support of an Advance Queensland Industry Research Fellowship and an Australian Research Council Industry Fellowship (IE230100245). S.H. acknowledges the support of ARC Industry Fellowship (IM230100030).

## Author contributions

Z.H., M.Z., S.H. and Z.Y. conceived the idea. Z.H. and X.C. conducted the experiments. Z.H. and Z.Y analysed data. L.L. and X.W. performed the LCA. Y.S. and K.X. conducted physical analysis. Z.H., M.Z. and Z.Y. wrote the manuscript with the input from all co-authors.

## Competing interests

The University of Queensland filed an International (PCT) Patent Application (No. PCT/AU2023/050697, Biogas Conversion Process) on 27 July 2023, partly based on the concept and data presented in this paper. Four of the authors of this paper, namely Z.Y, Z.H, M. Z. and S.H., who were/ are employees of The University of Queensland, are inventors of the patent. The patent is currently under review. The remaining authors declare no competing interests.
