## [Peer Review File · Nature Communications]

Integrated urban water management by coupling iron salt production and application with biogas upgradingREVIEWER COMMENTS

Reviewer #1 (Remarks to the Author):

The concept is interesting, novel (as far as I can tell) and the paper is well written. I am not absolutely sure about some technological aspects of the electrochemical reactor, which seems to be a bit complicated due to the mixing between the gas and aqueous phases, but I agree that this should not prevent the publication of this concept-based manuscript.

There are some aspects in the implementation, though, that need to be written more carefully, in my opinion. FeCO_3 is described in the paper as a remedy to almost all wastewater conveyance and treatment issues. If this was true, it would have been used despite its relatively high cost, and this is not the case in reality. First, normally ferric based salts are used for H_2S removal because they both oxidize the sulfide and the remaining ferrous precipitates with it (i.e., a double action). Second (and not secondly - line 30), ferric ion is also used for P removal because the K_{sp} values of ferric based phosphate solids are much lower than those of ferrous-based chemicals. Third, I am not sure about the advantage of increasing the alkalinity value due to the ferrous carbonate addition. This is negated by the precipitation of $\text{Fe}(\text{OH})_3$ that will occur after the ferrous oxidation to ferric and also, nitrification is seldom impeded by the lack of alkalinity, because in most plants denitrification compensates for most of the alkalinity that is lost in the nitrification (note that the alkalinity in WW is higher than in drinking water mostly because of the NH_3 that is released upon the hydrolysis of urea in the water). All in all, I think that the concept is worth publication but that the authors should play down a bit on the advantages and remind the reader of the possible disadvantages of the proposed concept, that include, inter alia, also the generation of much more sludge mass.

Reviewer #2 (Remarks to the Author):

Reviewer comments to manuscript NCOMMS-23-16547:

The manuscript titled "Integrated urban water management by coupling iron salt production and application with biogas upgrading" presents an integrated wastewater management scheme where a simple electrochemical system is used to upgrade biogas originating from anaerobically digested sewage sludge while simultaneously producing an iron salt (FeCO_3) to be used for various control purposes within the sewer system and treatment plant. An extensive set of laboratory experiments is complimented with a rough financial and life-cycle analysis to thoroughly study the effects of the proposed approach on wastewater management.

The key findings are that the electrochemical system is effective in utilising the CO_2 in biogas for the on-site generation of FeCO_3 , which simultaneously upgrades the biogas by reducing its CO_2 content from ca. 40% to <10%. A mass balance analysis also shows that the FeCO_3 production through this method could match the iron salt needs of the wastewater catchment area. The produced FeCO_3 can be used for different control measures in various parts of a sewer system and wastewater treatment plant, shown to facilitate (almost) complete removal of sulfide from anaerobic wastewater (in sewer) and in an anaerobic sludge digester when dosed at high enough concentrations. FeCO_3 addition to the activated sludge process also removes most of the phosphate from wastewater, thereby improving the effluent water quality and mitigating eutrophication in receiving water bodies. In addition to solely looking at the effect of FeCO_3 on these individual unit processes, the study also assesses the flow-on effects of FeCO_3 by dosing it only into the anaerobic wastewater (in sewer) and using this wastewater for consequent downstream processes, as would happen in real life. The results show that FeCO_3 dosage into sewage has beneficial flow-on effects all the way to the anaerobic digestion of sewage sludge.

In my view, the study is a great contribution towards a more circular and resource recovery focused approach into wastewater management and could be highly beneficial for the field.

The set of laboratory experiments carried out within the study seems meticulously planned to

answer a wide range of individual research objectives within the scope of the study. The experiments were carried out in triplicates for increased reliability and statistical significance, which is a good practice. Control experiments without iron dosing were included and statistical analysis was used to identify statistically significant differences between control experiments and experiments involving FeCO₃ dosage. The manuscript is well-structured and clearly written, which makes it easy to follow and understand.

To improve the manuscript even further, I have a few comments/suggestions:

Major comments:

1. In the abstract (lines 20-23), it is stated that the FeCO₃ produced and used in the study led to comparable or superior sulfide and phosphate removal and sludge settleability and dewaterability enhancement when compared to other, more commonly used iron salts. In my view, however, the manuscript does not provide enough support for this claim. If the superior performance of FeCO₃ over other iron salts is one of the main findings the authors want to highlight in the abstract, I feel like it needs to be backed up with more evidence:

- a. The effect of FeCO₃ dosing on, e.g., phosphate and sulfide removal was experimentally compared with a control with no iron dosing. Ideally, there would have been another control in which the iron was dosed as a different type of salt (e.g., FeCl₂) under otherwise similar operational conditions, which would have facilitated the direct comparison of FeCO₃ and FeCl₂ performance in the reactors. If possible, I would consider additional experiments using a commercial, commonly used iron salt for comparative purposes.
- b. If additional control experiments are not possible, the Results and Discussion section at least needs additional comparison of the results obtained with FeCO₃ to literature values reported for other iron salts. Currently, the performance of FeCO₃ is only compared to other iron salts in regard to sulfide removal from anaerobic sewage (lines 152-153 and 157-158). Also here, the second comparison (lines 157-158) is rather vague and would benefit from the addition of numerical values.
- c. No comparison to other iron salts was made when discussing the effect of FeCO₃ on phosphate removal or sludge settleability and dewaterability. Literature references should therefore be added to these sections (i.e., lines 163-170 and 196-199).

2. On lines 228-229, it is briefly mentioned that the proposed scenario is likely profitable since the electrochemical cell is simple to build and operate. I feel like it would be beneficial for the discussion to elaborate on this. The laboratory experiments were all carried out in short-term batch tests. For longer-term operation at a real wastewater treatment facility, what kind of challenges/changes could be expected for the electrochemical system? How often would the sacrificial iron electrodes need to be changed? How about the electrolyte?

Minor comments:

3. With such a large number of experiments aiming to provide answers to different objectives, I feel like the reader could benefit from a visual/table presentation summarising all the different laboratory experiments. This way, it would be easy to get an idea of the full scope and experimental structure of the work at a glance.

4. On line 355, the authors refer to the "theoretical reaction stoichiometry" but do not provide the equation for the reaction between iron and sulfide. It would be good to add the reaction equation here, similarly to the electrochemical reaction equations included in the Results and Discussion section.

5. As a general reminder, I want to highlight the importance of using colourblind-safe colour schemes for all figures to improve the accessibility of the manuscript. I therefore want to encourage the authors to consider this if they haven't already. Please ignore the comment if accessible colour schemes are already in place. (I am not colourblind myself so I can't provide any first-hand feedback on the perception of the chosen colours.)

Reviewer #3 (Remarks to the Author):

This manuscript reports a water treatment system by interesting different well-known processes. One of the major objectives was to replace the existing iron salt sources for water treatment using FeCO_3 which was generated electrochemically. The authors expect this process could establish local and environmentally friendly iron salt supplies, but still, where was the Fe^{2+} coming from and where did it end with? The reviewer is not convinced that this method could provide a solid solution to the existing problem. Overall, the reviewer has difficulty seeing the sufficient novelty of this manuscript for being published in this prestigious journal, and the results obtained contribute only incremental knowledge to the field of research.

Specific comments

1. Line 59-60: The authors mentioned biogas produced at WWTP is currently used for thermal and electricity. However, references 22-24 are from 2013-2018. Is there some literature published recently? Also, the reference is not very relevant. Why the obtained value is low?
2. Line 65: What kinds of left-behind materials would be produced? Which could result in secondary pollution.
3. Line 87-89: How the authors confirm that the dissolved inorganic carbon could be removed completely.
4. Line 135-138: What's the purpose of the measurement of the particle size distribution?
5. Line 145: Did the authors calculate what was the percentage of FeCO_3 ? A dominant iron compound is not good enough.
6. Line 146: How could the application of FeCO_3 affect sludge management? The management is not only the settleability and dewaterability, it could also be the change of the sludge amount, treatment, and disposal of the sludge.
7. Line 213-217: If the presence of the COD could affect the performance of FeCO_3 ?
8. Line 236: Is the value of 1000 km and 4000 km the normal transport value?
9. Line 262: Is the residual of H_2 a waste of electricity?
10. Line 280: How much biogas could be treated with two iron plates? How often do you need to change the iron plates?
11. Line 307-309: which reaction limits the experiment process? Fe dissolved or CO_2 capture? How does the author confirm the supplied Fe dissolve speed is equal to the supplied CO_2 ?
12. Line 392-393: The TS of both activated sludge and inoculated digested sludge should be considered.

Reviewer #1 (Remarks to the Author):

The concept is interesting, novel (as far as I can tell) and the paper is well written. I am not absolutely sure about some technological aspects of the electrochemical reactor, which seems to be a bit complicated due to the mixing between the gas and aqueous phases, but I agree that this should not prevent the publication of this concept-based manuscript.

Response: Thanks for your positive comments! The reactor design is expected to be very simple. While engineering details are not the focus on this proof-of-concept paper, we expect that biogas will be provided via a gas distribution system generating microbubble in a relatively tall reactor. CO₂ in the up-travelling gas bubbles will diffuse into the liquid phase and dissociate as bicarbonate and carbonate immediately in the alkali solution. The carbonate thus formed will react with the electrochemically generated Fe²⁺ in the solution forming FeCO₃, which settles to the bottom of the reactor for extraction.

We have added the following text to highlight that detailed engineering designs need to be further investigated:

Line 387-397: Numerous engineering aspects require further research in the upscaling of the process. In this proof-of-concept study, biogas was distributed to a short reactor using a 0.5 mm diameter needle. In full-scale applications, we expect that the biogas will be provided via a gas distribution system generating microbubbles in a relatively tall reactor. The reactor should be designed such that an adequate gas retention time is achieved for CO₂ in the up-travelling gas bubbles to diffuse into the liquid phase and dissociate as bicarbonate and carbonate, before the gas bubbles reach the reactor headspace releasing CH₄ and H₂. The electrode should be designed such that Fe²⁺ can be produced at a rate required for CO₂ removal with an acceptable voltage and power consumption. Also, experiments should be performed at full-scale sewer networks, wastewater treatment plants, and anaerobic sludge digesters, operated in a variety of conditions, to test the effectiveness of the E-FeCO₃ slurry for the intended purposes.

There are some aspects in the implementation, though, that need to be written more carefully, in my opinion. FeCO₃ is described in the paper as a remedy to almost all wastewater conveyance and treatment issues. If this was true, it would have been used despite its relatively high cost, and this is not the case in reality.

Response: As the reviewer highlighted, FeCO₃ has indeed not been applied to wastewater management. The industry survey conducted in 2011 in Australia regarding in-sewer chemical dosing for sulfide control and a more recent comprehensive literature review both confirmed this^{1,2}.

The truth is that the commercially available FeCO₃ (C-FeCO₃) does not have the same effects as our freshly produced FeCO₃ slurry. In the revised manuscript, we have included results we obtained by adding C-FeCO₃ to wastewater, wastewater sludge, and anaerobic sludge digesters, which showed C-FeCO₃ lacks the ability to remove sulfide or phosphate or to improve sludge settleability or dewaterability. The physical characteristics assessment of the C-FeCO₃ and E-

FeCO₃ revealed that the former has a more stable crystallised structure in large particles, limiting its reaction rate with sulfide and phosphate ions in the wastewater or sludge.

We have added the following text to elaborate the results of our additional experiments:

Line 355-362: This study proposed and experimentally demonstrated, for the first time, that E-FeCO₃, despite in a solid form as particles, can replace soluble iron salts for wastewater management. However, the industry survey conducted in Australia and a more recent comprehensive literature review both showed the C-FeCO₃ had not been applied to wastewater and sludge management^{1,2}. Our comparative experiments showed that the C-FeCO₃ is not effective in sulfide or phosphate precipitation (Fig. 5), due to its stable crystallised structure in larger particles (Supplementary Fig. 14). The low reactivity could explain why the C-FeCO₃ has not found applications in urban water management.

First, normally ferric based salts are used for H₂S removal because they both oxidize the sulfide and the remaining ferrous precipitates with it (i.e., a double action).

Response: It is true that Fe³⁺ removes sulfide via two sequential reactions, sulfide oxidation ($2\text{Fe}^{3+} + \text{HS}^- \rightarrow 2\text{Fe}^{2+} + \text{S}_{(\text{s})}^0 \downarrow$), followed by Fe²⁺ precipitation with additional sulfide ($\text{Fe}^{2+} + \text{S}^{2-} \rightarrow \text{FeS} \downarrow$). The reaction stoichiometry thus obtained (1.5 mole sulfide per mole of Fe) is more favourable than that obtained with Fe²⁺ addition (1:1). However, it needs to be noted that, under pH-neutral conditions (typical for domestic wastewater), Fe²⁺ is highly selective for sulfide. In contrast, Fe³⁺ is much less specific for sulfide, as it precipitates with other anions such as phosphate and hydroxide, and also coagulates with particulate and colloidal organic matters. Therefore, the use of Fe³⁺ does not necessarily lead to better sulfide removal. Also, the oxidation-reduction reaction between Fe³⁺ and sulfide is much slower than the precipitation reaction between Fe²⁺ and sulfide. Consequently, the addition of Fe²⁺ could remove sulfide more quickly.

Another factor determining their use is availability. FeCl₂ is often obtained as a direct by-product of the steel washing process in steel manufacturing. The production of FeCl₃ from FeCl₂ requires an additional oxidation process with e.g. chlorine.

As the paper does not focus on the comparison between Fe²⁺ and Fe³⁺, we have opted not to add the above discussion in the manuscript to avoid distracting readers.

Second (and not secondly - line 30), ferric ion is also used for P removal because the K_{sp} values of ferric based phosphate solids are much lower than those of ferrous-based chemicals.

Response: This is true. Fe³⁺ is sometimes added to the primary settling tank in a wastewater treatment plant, or to the secondary effluent, to remove phosphate. This cannot be replaced by Fe²⁺-based salts including E-FeCO₃. However, E-FeCO₃ (indeed any other Fe²⁺-based salts as well) could be added to the aeration basin to remove phosphate, as demonstrated in this paper.

We added the following text to clarify:

Line 378-381: Fe^{3+} salts are sometimes dosed to the primary settling tank in a wastewater treatment plant, or to the secondary effluent to remove phosphate. These cannot be replaced with Fe^{2+} -based salts including E- FeCO_3 . However, E- FeCO_3 (indeed any other Fe^{2+} -based salts as well) could be added to the aeration basin to remove phosphate (Fig. 2).

Third, I am not sure about the advantage of increasing the alkalinity value due to the ferrous carbonate addition. This is negated by the precipitation of $\text{Fe}(\text{OH})_3$ that will occur after the ferrous oxidation to ferric and also,

Response: It is true Fe^{3+} (added or formed via Fe^{2+} oxidation) precipitates with hydroxide, leading to consumption of alkalinity. In fact, reactions between Fe^{3+} or Fe^{2+} and other anions such as sulfide or phosphate would also consume alkalinity. For example, in pH-neutral condition, if 1 mole sulfide precipitates with Fe^{2+} , about 1.5 mole proton will be released ($\text{Fe}^{2+} + \text{H}_2\text{S} \rightarrow \text{FeS} \downarrow + 2\text{H}^+$; $\text{HS}^- \leftrightarrow \text{S}^{2-} + \text{H}^+$). Similarly, the precipitation of 1 mole phosphate with Fe^{3+} as FePO_4 will also release about 1.5 mole proton ($\text{H}_2\text{PO}_4^- \leftrightarrow \text{HPO}_4^{2-} + \text{H}^+$; $\text{HPO}_4^{2-} \leftrightarrow \text{PO}_4^{3-} + \text{H}^+$). Therefore, the use of Fe^{2+} and Fe^{3+} will reduce the alkalinity.

If the counter ion for Fe^{2+} and Fe^{3+} is Cl^- , pH in sewage will drop as a result of the above alkalinity consumption.

This will not be the case with E- FeCO_3 . The counter ion in this case is CO_3^{2-} , which forms bicarbonate and CO_2 , with the consumption of H^+ , under pH-neutral conditions. This negates the alkalinity consumption by the $\text{Fe}^{2+}/\text{Fe}^{3+}$ -induced precipitation reactions, thus preventing pH from decreasing that would otherwise occur. Our experimental results demonstrated that pH was actually raised slightly.

The slight pH increase achieved with E- FeCO_3 addition is certainly favourable over the pH drop caused by FeCl_2 addition. A higher pH means that a lower fraction of the dissolved sulfide will be in the form of H_2S . This would reduce the liquid to gas H_2S transfer.

We have undertaken additional experiments to compare E- FeCO_3 with other salts (i.e., C- FeCO_3 , FeCl_2 , and FeCl_3) in their wastewater and sludge management performance. These additional experiments show that the dose of E- FeCO_3 increases the pH of wastewater and sludge system, while the use of FeCl_2 and FeCl_3 decreases the pH.

We have added the following text to further explain the importance of the additional alkalinity added with E- FeCO_3 for sewer sulfide control:

Line 363-370: Compared to FeCl_2 and FeCl_3 , the dosage of E- FeCO_3 to wastewater or sludge causes a slight rise rather than a drop of pH, as in the case of FeCl_2 and FeCl_3 dosing. This is because the dosage of E- FeCO_3 provides additional alkalinity. In comparison, the dosage of FeCl_2 and FeCl_3 consumes alkalinity. In sewer networks, an increase of sewage pH is desirable as it shifts the H_2S and HS^- equilibrium towards HS^- ($\text{H}_2\text{S} \leftrightarrow \text{HS}^- + \text{H}^+$), and hence reduces the transfer of H_2S from the liquid to the gas phase. In fact, alkali in the form of e.g. $\text{Mg}(\text{OH})_2$ is often added to sewage in areas where H_2S is problematic ^{1,2}, clearly illustrating the importance of pH elevation for H_2S control in sewers.

nitrification is seldom impeded by the lack of alkalinity, because in most plants denitrification compensates for most of the alkalinity that is lost in the nitrification (note that the alkalinity in WW is higher than in drinking water mostly because of the NH_3 that is released upon the hydrolysis of urea in the water).

Response: Indeed, denitrification regenerates alkalinity, which compensates for the alkalinity consumption by nitrification. However, this compensation is partial. One mole alkalinity (as CaCO_3) is consumed per mole of ammonium oxidised. In comparison, only 0.5 mole of alkalinity is regenerated when one mole of nitrate or nitrite is reduced. Assuming 80% nitrogen removal via nitrification and denitrification, the amount of alkalinity regenerated is only 40% of the alkalinity consumed by nitrification. Therefore, the alkalinity to ammonium ratio in wastewater is still important for plants performing nitrification and denitrification. Normally a ratio of 0.6 mole alkalinity- CaCO_3 per mole ammonia nitrogen or higher is needed. However, this ratio is not always met.

New text has been added to highlight the importance of the additional alkalinity added with E- FeCO_3 for nitrification.

Line 370-375: In the wastewater treatment process, the additional alkalinity provided via E- FeCO_3 dosage is potentially beneficial for nitrification, particularly for wastewaters containing relatively low levels of alkalinity. Although denitrification partially regenerates alkalinity consumed by nitrification, wastewater in many parts of the world still does not contain alkalinity at a level enabling satisfactory nitrogen removal^{3,4}. In such cases, E- FeCO_3 should be a better source of iron than FeCl_2 or FeCl_3 .

All in all, I think that the concept is worth publication but that the authors should play down a bit on the advantages and remind the reader of the possible disadvantages of the proposed concept, that include, inter alia, also the generation of much more sludge mass.

Response: We hope that we have adequately addressed the reviewer's main concerns. We agree that, for this concept, further research is still needed before full-scale application. For example, we need to identify the optimal electrochemical cell configuration, including plate spacing, bubble size, and reactor size. Also, compared to the use of other iron salts, one potential disadvantage is that E-FeCO₃ is a slurry, which could settle in sewers. Although we have demonstrated this will unlikely be a problem via laboratory experiments (Supplementary Figure 8), this possibility should be further evaluated in practical applications.

We have discussed the potential future research in response to your first comment. We refer to our response therein.

It is true that the addition of iron salts increases sludge production, but this is not unique for E-FeCO₃. In an era of resource recovery, the production of more primary sludge is favourable for bioenergy recovery. In addition, as demonstrated in this as well as our previous work, the addition of iron salts helps improve sludge settleability and dewaterability, which means additional sludge production does not lead to increased sludge disposal costs.

We have added the following text in response to the reviewer's comment:

Line 382-386: The majority of iron dosed to the wastewater or the wastewater sludge will end in the biosolids as solid iron salts including iron-phosphate compounds, e.g. Fe₃(PO₄)₂ (vivianite)⁵, which increases sludge production. However, as demonstrated in this as well as previous work, the addition of iron salts helps improve sludge settleability and dewaterability, which means additional sludge production may not necessarily lead to increased sludge disposal costs.

Reviewer #2 (Remarks to the Author):

Reviewer comments to manuscript NCOMMS-23-16547:

The manuscript titled “Integrated urban water management by coupling iron salt production and application with biogas upgrading” presents an integrated wastewater management scheme where a simple electrochemical system is used to upgrade biogas originating from anaerobically digested sewage sludge while simultaneously producing an iron salt (FeCO_3) to be used for various control purposes within the sewer system and treatment plant. An extensive set of laboratory experiments is complimented with a rough financial and life-cycle analysis to thoroughly study the effects of the proposed approach on wastewater management.

The key findings are that the electrochemical system is effective in utilising the CO_2 in biogas for the on-site generation of FeCO_3 , which simultaneously upgrades the biogas by reducing its CO_2 content from ca. 40% to <10%. A mass balance analysis also shows that the FeCO_3 production through this method could match the iron salt needs of the wastewater catchment area. The produced FeCO_3 can be used for different control measures in various parts of a sewer system and wastewater treatment plant, shown to facilitate (almost) complete removal of sulfide from anaerobic wastewater (in sewer) and in an anaerobic sludge digester when dosed at high enough concentrations. FeCO_3 addition to the activated sludge process also removes most of the phosphate from wastewater, thereby improving the effluent water quality and mitigating eutrophication in receiving water bodies. In addition to solely looking at the effect of FeCO_3 on these individual unit processes, the study also assesses the flow-on effects of FeCO_3 by dosing it only into the anaerobic wastewater (in sewer) and using this wastewater for consequent downstream processes, as would happen in real life. The results show that FeCO_3 dosage into sewage has beneficial flow-on effects all the way to the anaerobic digestion of sewage sludge.

In my view, the study is a great contribution towards a more circular and resource recovery focused approach into wastewater management and could be highly beneficial for the field.

The set of laboratory experiments carried out within the study seems meticulously planned to answer a wide range of individual research objectives within the scope of the study. The experiments were carried out in triplicates for increased reliability and statistical significance, which is a good practice. Control experiments without iron dosing were included and statistical analysis was used to identify statistically significant differences between control experiments and experiments involving FeCO_3 dosage. The manuscript is well-structured and clearly written, which makes it easy to follow and understand.

Response: We appreciate these highly positive comments!

To improve the manuscript even further, I have a few comments/suggestions:

Major comments:

1. In the abstract (lines 20-23), it is stated that the FeCO₃ produced and used in the study led to comparable or superior sulfide and phosphate removal and sludge settleability and dewaterability enhancement when compared to other, more commonly used iron salts. In my view, however, the manuscript does not provide enough support for this claim. If the superior performance of FeCO₃ over other iron salts is one of the main findings the authors want to highlight in the abstract, I feel like it needs to be backed up with more evidence:

a. The effect of FeCO₃ dosing on, e.g., phosphate and sulfide removal was experimentally compared with a control with no iron dosing. Ideally, there would have been another control in which the iron was dosed as a different type of salt (e.g., FeCl₂) under otherwise similar operational conditions, which would have facilitated the direct comparison of FeCO₃ and FeCl₂ performance in the reactors. If possible, I would consider additional experiments using a commercial, commonly used iron salt for comparative purposes.

Response: We agree with this comment. We previously opted to compare our results with well-established literature values with FeCl₂, but we agree that it is scientifically more rigorous to compare results from parallel experiments.

We have performed all additional experiments suggested by the reviewer. The results support all our conclusions previously established. These new experiments have enhanced the robustness of our conclusions. Additionally, these new experiments also clearly show that, unlike the E-FeCO₃, the C-FeCO₃ does not have the ability to remove sulfide or phosphate from wastewater/sludge, or to enhance sludge settleability or dewaterability. This is attributed to the highly crystalized structure of the C-FeCO₃. These new results are reported in the revised manuscript and in the revised Supplementary Information, through the addition of the following sections:

Line 212-253:

Comparison of E-FeCO₃ with other iron salts in wastewater and sludge management performance

The experimental results provided compelling evidence for the efficacy of E-FeCO₃ in removing sulfide and phosphate, as well as its ability to improve sludge settleability and dewaterability. Further, the performance of E-FeCO₃ was compared with the commercially available FeCO₃ (C-FeCO₃), FeCl₂, and FeCl₃. C-FeCO₃ displayed negligible ability to remove sulfide or phosphate from wastewater/sludge, and limited ability to improve sludge settleability and dewaterability (Fig. 5). This could be attributed to its more stable crystalized structure in

larger particles (Supplementary Fig. 14), which possibly reduced its reaction rate with sulfide and phosphate ions in the wastewater and sludge.

In contrast, E-FeCO₃, FeCl₂, and FeCl₃ were all effective in eliminating sulfide and phosphate and in enhancing sludge settleability and dewaterability (Fig. 5). Specifically, in anaerobic sewage, the dissolved sulfide control efficiencies of E-FeCO₃, FeCl₂, and FeCl₃ were 0.53 ± 0.02 g S/g Fe, 0.51 ± 0.02 g S/g Fe, and 0.56 ± 0.03 g S/g Fe, respectively (Fig. 5a). Dissolved sulfide concentrations below 0.1 mg S/L were achieved in the overdosing tests with all these iron salts (Supplementary Fig. 15a). Similarly, in the anaerobic sludge digesters, all three iron salts reduced the dissolved sulfide and gaseous sulfide concentrations to below 2 mg S/L and 200 ppmv, respectively, with nearly 90% reduction. (Fig. 5f). The methane production and sulfate reduction processes were not impacted (Supplementary Fig. 15e and f). E-FeCO₃, FeCl₂, and FeCl₃ also displayed similar efficiencies in phosphate removal from aerated sludge, at 0.47 ± 0.02 g S/g Fe, 0.51 ± 0.02 g S/g Fe, and 0.52 ± 0.02 g S/g Fe, respectively (Fig. 5c). Furthermore, the uses of E-FeCO₃, FeCl₂, and FeCl₃ increased the sludge settleability by $36.9 \pm 6.2\%$, $37.4 \pm 3.4\%$, and $49.7 \pm 3.6\%$, respectively, and enhanced the sludge dewaterability by $55.0 \pm 1.5\%$, $57.9 \pm 5.3\%$, and $63.8 \pm 3.2\%$, respectively (Fig. 5h). Overall, the performance of E-FeCO₃ and FeCl₂ in sulfide and phosphate removal and in improving sludge settleability and dewaterability is remarkably similar, which is similar to, or slightly lower than that of FeCl₃. Fe³⁺ is able to oxidise sulfide (in addition to Fe²⁺ and sulfide precipitation) and is also known to have stronger flocculating or coagulating capabilities compared to Fe²⁺, which may explain the performance difference observed. The performance of FeCl₂ and FeCl₃ observed in these tests is comparable to that reported in literature (Supplementary Table 2), supporting the reliability of the results reported herein.

In contrast, the doses of E-FeCO₃, FeCl₂, and FeCl₃ induced different pH shifts. Dosages of ~30 mg Fe/L of E-FeCO₃, FeCl₂, and FeCl₃ to anaerobic sewage altered sewage pH by 0.15, -0.02 and -0.13 units, respectively (Fig. 5b and Supplementary Fig. 15b). A higher dose at ~90 mg Fe/L caused pH variations of 0.31, -0.49 and -0.66 units, respectively (Fig. 5b and Supplementary Fig. 15b). Similar pH variation patterns were also observed in the experiments with activated sludge (Fig. 5d and Supplementary Fig. 15d) and anaerobic sludge digesters (Fig. 5g). In all these cases, the provision of additional alkalinity via E-FeCO₃ addition, in comparison to the consumption of alkalinity via FeCl₂ or FeCl₃ dosage, is favourable, as will be further discussed later (Discussion Section).

In conclusion, E-FeCO₃ proves to be a suitable replacement for FeCl₂ and FeCl₃ for wastewater or wastewater sludge management, while C-FeCO₃ shows limited effectiveness.

Fig. 5. Comparison of E-FeCO₃ with other iron salts in wastewater and sludge management performance. a, Sulfide control in sewer (underdosing). **b**, pH at the beginning and end of each test in sewer (underdosing). **c**, Phosphate removal in an aerated activated sludge (underdosing). **d**, pH at the beginning and end of each test in an aerated activated sludge (underdosing). **e**, Dissolved sulfide control in an anaerobic sludge digester. **f**, Gaseous sulfide control in an anaerobic sludge digester. **g**, pH at the beginning and end of each test in an anaerobic sludge digester. **h**, Improvement to sludge settleability and dewaterability. All values are means \pm standard deviations of triplicate tests.

Fig. 14 | The characteristic of fresh C-FeCO₃ and E-FeCO₃ (produced in experiments at pH 8.5). a, The particle size distribution. b, Values of D₁₀, D₅₀, and D₉₀. c, XRD profile of the FeCO₃ of C-FeCO₃. d, XRD profile of the FeCO₃ of E-FeCO₃. e, SEM micrograph of the C-FeCO₃. f, SEM micrograph of the E-FeCO₃. All values are means ± standard deviations of triplicate tests.

Fig. 15| Application of different iron salts to wastewater and sludge management. a, Sulfide control in sewer (overdosing). **b,** pH at the beginning and end of each test in sewer (overdosing). **c,** Phosphate removal in an aerated activated sludge (overdosing). **d,** pH at the beginning and end of each test in an aerated activated sludge (overdosing). **e,** Methane production in BMP tests. **f,** Sulfate reduction in BMP tests. Averages of triplicate experiments are reported, with error bars representing standard deviations.

Supplementary Table 2. The use of different iron salts in urban wastewater management systems.

	Iron source	Iron dosage	Dosing point	Efficiency	Reference
Dissolved sulfide control in sewer					
	FeSO ₄	1.47 g Fe/g S	Sewer network	Sulfide removal capacity: 0.59 g S/g Fe	6
	FeClSO ₄	1.19 g Fe/g S		Sulfide removal capacity: 0.74 g S/g Fe	
	Ferrous salt	2.28 g Fe/g S	Sewer network	Reducing the dissolved sulfide concentration to <0.1 mg S/L	7
	Ferric salt	1.58 g Fe/g S			
	FeCl ₃	10 mg Fe/L-wastewater	Sewer network	Sulfide removal capacity: 0.43 g S/g Fe	8
	FeCl ₃	21 mg Fe/L-wastewater	Sewer network	Reducing the dissolved sulfide concentration from 17.1 to <0.2 mg S/L	9
	FeCl ₃	49 mg Fe/L-wastewater	Sewer network	Sulfide removal capacity: 0.42 g S/g Fe	10
Phosphate control in wastewater treatment system					
		0.57 molar Fe/molar P	Wastewater treatment system	P removal capacity: ~0.49 g P/g Fe	11
	FeCl ₃	1.08 molar Fe/molar P		P removal capacity: ~0.36 g P/g Fe	
		1.48 molar Fe/molar P		P removal capacity: ~0.30 g P/g Fe	
	Fe(OH) ₃	2.85 molar Fe/molar P		P removal capacity: ~0.09 g P/g Fe	

		3.86 molar Fe/molar P		P removal capacity: ~0.10 g P/g Fe	
		> 4.80 molar Fe/molar P		P removal capacity: <0.09 g P/g Fe	
	FeCl ₃	35 mg Fe/L-wastewater		Reducing the P concentration from ~6.7 to ~0.1 mg P/L	12
	FeCl ₃	20 mg Fe/L-wastewater		Reducing the P concentration from to ~7.1 to ~0.3 mg P/L	13
	FeCl ₂	20 mg Fe/L-wastewater	Sewer network (Evaluating the flow-on effect on wastewater treatment system)	P removal capacity: 0.37 g P/g Fe	14
	FeCl ₃	20 mg Fe/L-wastewater		P removal capacity: 0.44 g P/g Fe	
	FeCl ₃	10 mg Fe/L-wastewater	Sewer network (Evaluating the flow-on effect on wastewater treatment system)	P removal capacity: 0.47 g P/g Fe	8
Dissolved and gaseous sulfide control in anaerobic digestion					
	FeCl ₃	7.40–9.43 mg Fe/g TS		H ₂ S removal efficiency of ~93%	15
	FeCl ₃	4.5 mg Fe/g TS	Anaerobic digestion	H ₂ S removal efficiency of ~87%	16
	FeCl ₃	5–20 mg Fe/L-wastewater	Sewer network (Evaluating the flow-on effect on anaerobic digestion)	Reducing the dissolved sulfide concentration to <0.2 mg S/L	17
	FeCl ₃	10 mg Fe/L-wastewater	Sewer network (Evaluating the flow-on effect on anaerobic digestion)	 Reducing the dissolved sulfide concentration from ~23.7 to ~2.7 mg S/L (~88.5% removal efficiency) 	8

- Reducing the H₂S concentration in biogas from ~911 to ~130 ppmv (~82.3% removal efficiency)

Sludge settleability

FeCl ₃	10 mg Fe/L-wastewater	Sewer network (Evaluating the flow-on effect on wastewater treatment system)	Reducing the SVI from ~75 to ~55 mL/g	8
FeCl ₃	20 mg Fe/L-wastewater	Wastewater treatment system	Reducing the SVI from ~78 to ~45 mL/g	13
FeCl ₃	35 mg Fe/L-wastewater	Wastewater treatment system	Reducing the SVI from ~105 to ~45 mL/g	12

Sludge dewaterability

FeCl ₃	35 mg Fe/L-wastewater	Wastewater treatment system	Reducing the SRF from $\sim 2.1 \times 10^{13}$ to $\sim 5.0 \times 10^{12}$ m/kg	12
FeCl ₃	10 mg Fe/L-wastewater	Sewer network (Evaluating the flow-on effect on anaerobic digestion)	Increasing the sludge dewaterability from ~15.9 to ~19.4%	8
FeCl ₃	20 mg Fe/L-wastewater	Wastewater treatment system	Reducing the SRF from $\sim 2.3 \times 10^{13}$ to $\sim 3.4 \times 10^{12}$ m/kg	13

b. If additional control experiments are not possible, the Results and Discussion section at least needs additional comparison of the results obtained with FeCO_3 to literature values reported for other iron salts. Currently, the performance of FeCO_3 is only compared to other iron salts in regard to sulfide removal from anaerobic sewage (lines 152-153 and 157-158). Also here, the second comparison (lines 157-158) is rather vague and would benefit from the addition of numerical values.

Response: As detailed above, we have undertaken additional experiments to compare different iron salts under identical experimental conditions. The new results support all our previous conclusions.

c. No comparison to other iron salts was made when discussing the effect of FeCO_3 on phosphate removal or sludge settleability and dewaterability. Literature references should therefore be added to these sections (i.e., lines 163-170 and 196-199).

Response: Additional experiments have been performed. We refer to our response to a) and the changes quoted therein.

2. On lines 228-229, it is briefly mentioned that the proposed scenario is likely profitable when the concentration of dissolved Fe and carbonate reaches the threshold value since the electrochemical cell is simple to build and operate. I feel like it would be beneficial for the discussion to elaborate on this.

Response: We have added the following discussion to elaborate:

An electrochemical cell was implemented to achieve the multiple goals. The construction of this electrochemical cell is straightforward, utilizing iron plates as electrodes and a NaCl solution as the electrolyte, without requiring complex materials or membranes. The iron should ideally be sourced from locally recycled iron. If this is not available, it could be imported from iron manufactures. Even in the latter case, the transport costs would be greatly reduced. As an example, the FeCl_2 supplied in Australia is a solution containing 12% iron. Theoretically, transporting iron as iron plates instead of this iron salt solution would substantially reduce the transportation cost, with a ~88% reduction in weight and a ~98% reduction in volume. Furthermore, this electrochemical cell is easy to operate, and the control logic is simple. By adjusting the current to keep the electrolyte pH at a pre-selected level (recommended to be around 8.5), the amount of OH^- produced is ensured to just meets the demanded for CO_2 conversion to CO_3^{2-} and its subsequent removal as FeCO_3 .

The laboratory experiments were all carried out in short-term batch tests. For longer-term operation at a real wastewater treatment facility, what kind of challenges/changes could be expected for the electrochemical system? How often would the sacrificial iron electrodes need to be changed? How about the electrolyte?

Response: Our electrochemical process efficiently removes gaseous CO_2 by transforming it into the FeCO_3 slurry through three synchronous processes. Firstly, Fe is electrochemically

oxidized to Fe^{2+} . Secondly, CO_2 dissolved reacts with OH^- to form carbonate. Lastly, the E- FeCO_3 slurry is generated when the concentration of dissolved iron and carbonate reaches the threshold values determined by the FeCO_3 solubility product. In long-term operation at a real WWTP, the latter two processes are not expected to be rate-limiting steps. This is because the required mass transfer efficiency of this process is relatively low (the newly added Supplementary Table 4), and the precipitation process is usually instantaneous. Therefore, the potential bottleneck would likely be encountered in the first step during the upscaling process.

In the small laboratory reactor, two parallel iron plates served as the cathode and anode. However, for full-scale application, a challenge arises in determining how to construct the cathode and anode with multiple Fe plates. Taking the spacing between the Fe plates as an example, a large spacing would increase the reactor size and the operating voltage, leading to higher capital and operating costs. On the other hand, a small spacing may cause operating issues, such as short circuiting. Overall, the determination will involve conducting further research and experiments to identify the most suitable configuration for the large-scale electrochemical process.

The interval for electrode replacement is determined based on the reactor volume, plate spacing, and CO_2 loading rate. To estimate the replacement intervals, three scenarios are considered, representing different influent biodegradable COD (bCOD) concentrations, leading to different CO_2 loading rates (Supplementary Table 4). In a 200 m^3 reactor with an electrode spacing of 1 cm, the electrode replacement interval is estimated to be 302 days and 113 days, respectively, for the influent bCOD concentrations of 300 mg/L and 800 mg/L.

In the laboratory experiment, the electrolyte used is tap water with NaCl amendment at a concentration of 2 g/L. For full-scale applications, the plant effluent with a similar NaCl amendment can serve as the electrolyte. The cost for the electrolyte is negligible when compared to the overall input and output values of the process. The cost of NaCl has been added to the economic assessment.

We have added the following text in response to the reviewer’s comment:

Line 336-342: Serving as scarifying electrodes, the Fe plates need to be replaced regularly, with an interval determined by the reactor size, the plate spacing, and the CO_2 loading rate. In the example used in the input-output analysis, the CO_2 loading rate is estimated to be in the range of 656–1750 m^3/d , given a bCOD concentration range of 300–800 mg/L (Supplementary Table 4). Assuming a total electrochemical cell volume of 200 m^3 and an iron plate spacing of 1 cm, the Fe plate replacement interval is estimated to be 4–10 months (Supplementary Table 4), which is reasonable.

Supplementary Table 4. Estimation of the iron plate replacement interval time.

	Unit	Low carbon	Medium carbon	High carbon
Influent biodegradable COD concentration	g/m^3	300	500	800
Daily CO_2 production ¹	m^3/day	656	1094	1750

Daily Fe consumption ¹	kg Fe/day	1,470	2,450	3,920
Volume of electrochemical cell	m ³		200	
Required CO ₂ mass transfer efficiency (K _{La}) ²	/d	206	343	548
Iron plate spacing	cm		1	
Iron plate thickness	cm		1	
Volume of iron electrode when new ³	m ³		67	
Mass of electrode when new ⁴	kg		522,600	
Iron electrode replacement interval ⁵	day	302	181	113

¹See calculation in Supplementary Table 2. ²Assuming CO₂ accounts for 5% of the upgraded biogas. ³Assuming the volume of iron electrode is 1/3 of the total volume. ⁴Assuming iron density of 7800 kg/m³. ⁵Assuming replacement occurs at 85% consumption.

The potential future research for the full-scale application has been discussed in response to the first comment of Review 1. We refer to our response therein.

Minor comments:

With such a large number of experiments aiming to provide answers to different objectives, I feel like the reader could benefit from a visual/table presentation summarising all the different laboratory experiments. This way, it would be easy to get an idea of the full scope and experimental structure of the work at a glance.

Response: Accepting this good suggestion, we have summarised all experiments in a single figure presented as Supplementary Fig. 18. The new Figure is also quoted below.

Fig. 18 | The overall experimental design.

4. On line 355, the authors refer to the “theoretical reaction stoichiometry” but do not provide the equation for the reaction between iron and sulfide. It would be good to add the reaction equation here, similarly to the electrochemical reaction equations included in the Results and Discussion section.

Response: The reaction between iron and sulfide has been added.

5. As a general reminder, I want to highlight the importance of using colourblind-safe colour schemes for all figures to improve the accessibility of the manuscript. I therefore want to encourage the authors to consider this if they haven't already. Please ignore the comment if accessible colour schemes are already in place. (I am not colourblind myself so I can't provide any first-hand feedback on the perception of the chosen colours.)

Response: As suggested by the reviewer, all the figures in this manuscript has been updated using colourblind-safe colour schemes.

Reviewer #3 (Remarks to the Author):

This manuscript reports a water treatment system by interesting different well-known processes. One of the major objectives was to replace the existing iron salt sources for water treatment using FeCO_3 which was generated electrochemically. The authors expect this process could establish local and environmentally friendly iron salt supplies, but still, where was the Fe^{2+} coming from and where did it end with?

Response: With our system, the Fe^{2+} ions are produced by electrochemically dissolving iron plates. It is true that we still need a source of iron. The iron should ideally be sourced from locally recycled iron. If this is not available, it could be imported from iron manufactures. Even in the latter case, the transport costs would be greatly reduced. As an example, the FeCl_2 supplied in Australia is a solution containing 12% iron. Theoretically, transporting iron as iron plates instead of this iron salt solution would substantially reduce the transportation cost, with a ~88% reduction in weight and a ~98% reduction in volume.

The Fe^{2+} dosed to the wastewater or the wastewater sludge will end in the biosolids as iron-phosphate compounds e.g. $\text{Fe}_3(\text{PO}_4)_2$ (vivianite). The fate of Fe will depend on the fate of the biosolids in the local area, e.g. for land application or for incineration.

We have added the following text to answer the questions raised by the reviewer:

Line 326-331: The iron should ideally be sourced from locally recycled iron. If this is not available, it could be imported from iron manufactures. Even in the latter case, the transport costs would be greatly reduced. As an example, the FeCl_2 supplied in Australia is a solution containing 12% iron. Theoretically, transporting iron as iron plates instead of this iron salt solution would substantially reduce the transportation cost, with a ~88% reduction in weight and a ~98% reduction in volume.

Line 382-383: The majority of iron dosed to the wastewater or the wastewater sludge will end in the biosolids as iron-phosphate compounds, e.g. $\text{Fe}_3(\text{PO}_4)_2$ (vivianite) ⁵.

The reviewer is not convinced that this method could provide a solid solution to the existing problem. Overall, the reviewer has difficulty seeing the sufficient novelty of this manuscript for being published in this prestigious journal, and the results obtained contribute only incremental knowledge to the field of research.

Response: This study gives a powerful example how an out-of-the-box approach could lead to innovation enabling a circular economy. Biogas upgrading is seemingly unrelated to sewer infrastructure protection, nutrients removal and recovery from wastewater, and reduction in the sludge disposal costs. Hence the conventional physical, chemical and biological approaches to biogas upgrading have solely focused on the quality of the upgraded biogas and the associated costs. In the meantime, the same water utility has to invest in importing chemicals to protect its sewer infrastructure, and to improve the performance of its wastewater treatment plant. The synergy we have identified and demonstrated in this work between biogas upgrading and wastewater management enables water utilities to (1) kill two birds with one stone, namely to

significantly reduce its wastewater management costs while increasing the value of the bioenergy recovered, (2) recycle CO₂ produced in wastewater treatment to the urban water system to provide additional alkalinity as CO₃²⁻, along with the locally available recycle iron. The latter aspect also improves the supply chain security of iron salts.

Also importantly, we proposed and experimentally demonstrated, for the first time, that E-FeCO₃, despite in a solid form as particles, can replace soluble iron salts for wastewater management. Our industry survey¹ and a recent literature review² both showed that C-FeCO₃ has never been used as a source of iron for wastewater management. Indeed, our additional experiments (now reported in the revised paper) showed that C-FeCO₃ is not effective in sulfide or phosphate precipitation, due to its more stable crystallised structure in larger particles, in comparison to E-FeCO₃.

Finally, we would like to highlight that, since the completion of this laboratory work, we have secured over A\$2M funding to upscale the technology in the next few years, with support from major water utilities in Australia. This strongly suggests that the industry is convinced of the potential of this technology in addressing their problems.

We have amended the text to more clearly highlight the novelty and potential of the proposed method.

Line 316-323: We are entering an era of circular economy. This requires us to improve our traditional approaches to cater to the requirement of sustainable development. This study showcases an innovative technological solution within urban water management system. It establishes a unique connection between biogas upgrading and the enhancement of wastewater and sludge management. Specifically, it offers the potential to protect sewer infrastructure, facilitate the removal and recovery of nutrients from wastewater, and reduce costs associated with sludge disposal. The experimental findings demonstrated the feasibility of this out-of-the-box solution, highlighting its ability to address multiple challenges simultaneously.

Specific comments

1. Line 59-60: The authors mentioned biogas produced at WWTP is currently used for thermal and electricity. However, references 22-24 are from 2013-2018. Is there some literature published recently? Also, the reference is not very relevant. Why the obtained value is low?

Response: The use of biogas for electricity and/or heat energy production is still a standard industry practice. As requested, we have added more recent literature published in 2021 and 2022.

Biogas is typically utilized for generating electricity and/or heat within a WWTP. The value of the electricity generated is governed by the local electricity price, which is a low-value commodity in most parts of the world.

We have added the following text to answer the questions raised by the reviewer:

Line 61-64: However, limited by the relatively low electricity price and low value of thermal energy at most places, the value of biogas thus derived is generally low, especially considering the significant capital and maintenance costs associated with the gas engines ¹⁸.

2. Line 65: What kinds of left-behind materials would be produced? Which could result in secondary pollution.

Response: The following information has been added.

Line 67-70: However, they are often energy-inefficient and most leave behind materials requiring disposal or regeneration, potentially causing secondary pollution ¹⁹. For example, CO₂ absorption using amine solutions results in degraded solvent that are toxic to both humans and the environment ²⁰.

3. Line 87-89: How the authors confirm that the dissolved inorganic carbon could be removed completely.

Response: As no waste stream is included in our experimental design, the CO₂ in the biogas is either present in the upgraded biogas or in the FeCO₃ slurry. The experimental results showed that, in the steady state of our process at an optimal pH of 8.5, about 85% of CO₂ in the feed gas was retained in the reactor as FeCO₃, while the remaining 15% remained in the upgraded gas. The concentrations of dissolved CO₂, bicarbonate, and carbonate are calculated to be 2.02, 272.49, and 3.9 mM, respectively.

We have added the following text to answer the questions raised by the reviewer:

Line 348-350: In our proof-of-concept experiments at an optimal pH of 8.5, ~85% of CO₂ in the feed gas was retained in the reactor as FeCO₃, while the remaining ~15% remained in the upgraded gas.

4. Line 135-138: What's the purpose of the measurement of the particle size distribution?

Response: The FeCO₃ produced exists as solids in a slurry. Its efficacy in controlling sulfide in a real sewer network may be limited by the particle size, as excessively large particles might hinder their reactivity and/or their ability to remain suspended in the wastewater. Therefore, we needed to monitor the particle size distribution of the produced FeCO₃ slurry.

We have added the following text to answer the questions raised by the reviewer:

Line 144-146: The particle size was measured as they likely influence the efficacy of E-FeCO₃ to react with sulfide or phosphate when added to wastewater or sludge, due to e.g. surface limitations or solids settling.

5. Line 145: Did the authors calculate what was the percentage of FeCO₃? A dominant iron compound is not good enough.

Response: We have added the following text to estimate the percentage of FeCO₃ in the slurry.

Line 152-156: Three crystalline iron species in the E-FeCO₃ slurry were identified to be siderite (FeCO₃), goethite (α -FeO(OH)), and hematite (Fe₂O₃) (Supplementary Fig. 9). Among these, FeCO₃ is the only compound containing Fe²⁺, thus the measured fraction of Fe²⁺ in total Fe (86.2 ± 3.9%) represents the fraction of FeCO₃ in all Fe-containing compounds. This is consistent with the measured ratio between CO₂ removed and Fe oxidized ($R_{\text{CO}_2/\text{Fe}}$), which is 0.84 ± 0.03.

6. Line 146: How could the application of FeCO₃ affect sludge management? The management is not only the settleability and dewaterability, it could also be the change of the sludge amount, treatment, and disposal of the sludge.

Response: We have addressed a similar question raised by Review 1 (the last comment). We refer to our response therein.

7. Line 213-217: If the presence of the COD could affect the performance of FeCO₃?

Response: The wastewater COD concentration affects the amount of biogas produced, which subsequently determines the amount of E-FeCO₃ that can be produced. This means more E-FeCO₃ could be produced for wastewater with a higher COD concentration. Our mass balance analysis shows that an adequate amount of E-FeCO₃ could be produced for wastewater management even with a relatively low COD concentration (see Supplementary Table 2).

We have added the following text to answer this question:

Line 268-270: The wastewater biodegradable chemical oxygen demand (bCOD) concentration affects the amount of biogas produced, which subsequently determines the amount of E-FeCO₃ that can be produced.

8. Line 236: Is the value of 1000 km and 4000 km the normal transport value?

Response: Here we used the Australia data as examples. The typical transport distance ranges from several hundred to two thousand kilometres, so here we used 1000 km as an average for the current practice. A distance of 4000 km represents the possible longest transportation distance in Australia.

We have added the following text to explain:

Supplementary Table 5: The typical transport distance ranges from several hundred to two thousand kilometres, so here we used 1000 km as an average for the current practice. A distance of 4000 km represents the possible longest transportation distance in Australia.

9. Line 262: Is the residual of H₂ a waste of electricity?

Response: H₂, generally representing 35–40% of the upgraded biogas, contributes to the energy content of the upgraded biogas. It partially recovers the electricity energy invested, and is not a wasteful product.

We have added the following text to answer this question:

Line 352–354: CO₂ removed from the feed gas was replaced by H₂ generated at the cathode with a molar ratio of 1:1. The energy content of H₂ in the upgraded biogas partially recovers the electricity energy invested.

10. Line 280: How much biogas could be treated with two iron plates? How often do you need to change the iron plates?

Response: We have addressed a similar question raised by Reviewer 2 (the second major comment). We refer to our response therein.

11. Line 307-309: which reaction limits the experiment process? Fe dissolved or CO₂ capture? How does the author confirm the supplied Fe dissolve speed is equal to the supplied CO₂?

Response: Indeed, the CO₂ capture rate decides the maximum reaction rate of this process. The mass transfer efficiency of CO₂ from gas to liquid is determined by the gas retention time, the bubble size, operating pH, and reactor configuration. In comparison, the release rate of Fe²⁺ from the iron plate is determined by the current input, which is theoretically unrestricted.

We have added the following text to answer this question:

Line 343–348: The overall process comprises three key steps, namely the electrochemical production of Fe²⁺ and OH⁻, dissolution of CO₂ and its subsequent conversion to CO₃²⁻ under alkaline conditions, and the precipitation of Fe²⁺ and CO₃²⁻ as FeCO₃. Among these, the CO₂ transfer from the gas bubbles to the electrolyte is the rate-limiting step, which determines the rate of the overall electrochemical system. The CO₂ mass transfer rate is influenced by the reactor configuration, the gas flow rate, the gas bubble size, and the operating pH.

We control pH at a pre-selected pH set-point by adjusting the current. This ensures that the amount of Fe²⁺ produced matches the amount of CO₂ captured. We have added the following text to elaborate:

Line 332–335: Furthermore, this electrochemical cell is easy to operate, and the control logic is simple. By adjusting the current to keep the electrolyte pH at a pre-selected level (recommended to be around 8.5), the amount of OH⁻ produced is ensured to just meets the demanded for CO₂ conversion to CO₃²⁻ and its subsequent removal as FeCO₃.

12. Line 392-393: The TS of both activated sludge and inoculated digested sludge should be considered.

Response: The TS and VS concentrations of thickened activated sludge and inoculated digested sludge have been added.

Line 535–537: Briefly, about 20 mL thickened activated sludge (TS: 21.3 ± 0.1 g/L; VS: 17.7 ± 0.1 g/L) was mixed with ~40 mL inoculated digested sludge (TS: 20.6 ± 0.1 g/L; VS: 16.3 ± 0.1 g/L), and then transferred into a 100 mL sealed bottle.

Reference

1. Ganigue, R., Gutierrez, O., Rootsey, R. & Yuan, Z. Chemical dosing for sulfide control in Australia: an industry survey. *Water Res.* **45**, 6564-6574 (2011).
2. Cen, X., Li, J., Jiang, G. & Zheng, M. A critical review of chemical uses in urban sewer systems. *Water Res.*, 120108 (2023).
3. Lee, W., An, S. & Choi, Y. Ammonia harvesting via membrane gas extraction at moderately alkaline pH: A step toward net-profitable nitrogen recovery from domestic wastewater. *Chem. Eng. J.* **405**, 126662 (2021). <https://doi.org/https://doi.org/10.1016/j.cej.2020.126662>
4. Bodkhe, S. Y. A modified anaerobic baffled reactor for municipal wastewater treatment. *J. Environ. Manage.* **90**, 2488-2493 (2009). <https://doi.org/https://doi.org/10.1016/j.jenvman.2009.01.007>
5. Salehin, S. *et al.* Recovery of in-sewer dosed iron from digested sludge at downstream treatment plants and its reuse potential. *Water Res.* **174**, 115627 (2020).
6. Tomar, M. & Abdullah, T. H. A. Evaluation of chemicals to control the generation of malodorous hydrogen sulfide in waste water. *Water Res.* **28**, 2545-2552 (1994). [https://doi.org/https://doi.org/10.1016/0043-1354\(94\)90072-8](https://doi.org/https://doi.org/10.1016/0043-1354(94)90072-8)
7. Firer, D., Friedler, E. & Lahav, O. Control of sulfide in sewer systems by dosage of iron salts: Comparison between theoretical and experimental results, and practical implications. *Sci. Total Environ.* **392**, 145-156 (2008). <https://doi.org/https://doi.org/10.1016/j.scitotenv.2007.11.008>
8. Rebosura Jr, M. *et al.* A comprehensive laboratory assessment of the effects of sewer-dosed iron salts on wastewater treatment processes. *Water Res.* **146**, 109-117 (2018).
9. Zhang, L., Keller, J. & Yuan, Z. Inhibition of sulfate-reducing and methanogenic activities of anaerobic sewer biofilms by ferric iron dosing. *Water Res.* **43**, 4123-4132 (2009).
10. Cao, J., Zhang, L., Hong, J., Sun, J. & Jiang, F. Different ferric dosing strategies could result in different control mechanisms of sulfide and methane production in sediments of gravity sewers. *Water Res.* **164**, 114914 (2019).
11. Thistleton, J., Berry, T.-A., Pearce, P. & Parsons, S. Mechanisms of chemical phosphorus removal II: iron (III) salts. *Process Saf. Environ. Prot.* **80**, 265-269 (2002).
12. Hu, Z. *et al.* Centralized iron-dosing into returned sludge brings multifaceted benefits to wastewater management. *Water Res.* **203**, 117536 (2021). <https://doi.org/https://doi.org/10.1016/j.watres.2021.117536>
13. Hu, Z. *et al.* Novel Use of a Ferric Salt to Enhance Mainstream Nitrogen Removal from Anaerobically Pretreated Wastewater. *Environ. Sci. Technol.* **57**, 6712-6722 (2023).
14. Gutierrez, O., Park, D., Sharma, K. R. & Yuan, Z. Iron salts dosage for sulfide control in sewers induces chemical phosphorus removal during wastewater treatment. *Water Res.* **44**, 3467-3475 (2010). <https://doi.org/https://doi.org/10.1016/j.watres.2010.03.023>
15. Akgul, D., Abbott, T. & Eskicioglu, C. Assessing iron and aluminum-based coagulants for odour and pathogen reductions in sludge digesters and enhanced digestate dewaterability. **598**, 881-888 (2017).
16. Abbott, T. & Eskicioglu, C. Effects of metal salt addition on odor and process stability during the anaerobic digestion of municipal waste sludge. *Waste Manage.* **46**, 449-458 (2015). <https://doi.org/https://doi.org/10.1016/j.wasman.2015.07.050>
17. Ge, H., Zhang, L., Batstone, D. J., Keller, J. & Yuan, Z. Impact of iron salt dosage to sewers on downstream anaerobic sludge digesters: sulfide control and methane production. *J. Environ. Eng.* **139**, 594-601 (2013).
18. Pérez, V. c., Lebrero, R. & Muñoz, R. I. Comparative evaluation of biogas valorization into electricity/heat and poly (hydroxyalkanoates) in waste treatment plants: assessing the influence of local commodity prices and current biotechnological limitations. *ACS Sustainable Chemistry & Engineering* **8**, 7701-7709 (2020).
19. Angelidaki, I. *et al.* Biogas upgrading and utilization: Current status and perspectives. *Biotechnol. Adv.* **36**, 452-466 (2018). <https://doi.org/https://doi.org/10.1016/j.biotechadv.2018.01.011>
20. Hack, J., Maeda, N. & Meier, D. M. Review on CO₂ Capture Using Amine-Functionalized Materials. *ACS omega* **7**, 39520-39530 (2022).

REVIEWERS' COMMENTS

Reviewer #1 (Remarks to the Author):

My impression is that the authors have addressed most of the reviewers' concerns. I thought that the first version of the paper was adequate for publication and certainly did not change my mind after the revision.

Ori Lahav

Reviewer #2 (Remarks to the Author):

I want to wholeheartedly thank the authors for their efforts to improve the manuscript. I find that especially the additional control experiments carried out since the last revision round, rigorously comparing the performance of E-FeCO₃ with several different commercial iron salts (C-FeCO₃, FeCl₂ and FeCl₃), make the manuscript remarkably stronger. The control experiments show that the E-FeCO₃ produced in the electrochemical cell can achieve comparable results to the widely used FeCl₂ and FeCl₃ under identical operational conditions. Overall, the data set provided in the manuscript and the supplementary file is very comprehensive and supports the conclusions made in the article.

I also find the added Discussion section highly beneficial for the article. In my view, this section gives a good overview of the additional benefits obtainable when using E-FeCO₃ over FeCl₂ and FeCl₃, but also discusses the challenges of scaling up the E-FeCO₃ production system and identifies further research needs.

I am satisfied with how the authors addressed my (and the other reviewers') comments from the previous revision round and recommend this article to be published. As stated in my previous review, I think the proposed concept could be highly beneficial for the water treatment sector.

Reviewer #3 (Remarks to the Author):

Though it is a proof-of-concept study, the feasibility of a real case application should be considered. The whole process/system reported in this work is very complicated. Though the authors believe it is simple, still it is a combination of several processes. The authors should also provide a more detailed analysis and at least compare their methods with many other emerging and existing technologies in wastewater treatment. As the reviewer commented, if it is economically feasible, it should have been applied already. The concept is not new, as it has been proposed or at least discussed in the past. The references provided by the authors also confirmed that the concept is not new. The new information is that the authors tested the concept in the lab, but the results obtained open many uncertainties. The novelty and the quality of this work don't reach the level of being published in this journal. The reviewer would suggest the authors submit it to more technical journals with a more specific readership.

General response to reviewers' comments

We would like to sincerely thank all reviewers for thoroughly reviewing the revised paper. We are pleased that Reviewer #1 and Reviewer #2 are satisfied with our revisions. The further comments (in **black**) from Reviewer#3 are addressed below. Responses from the authors in **blue**, and the revisions are quoted in **red**.

Reviewer #3 (Remarks to the Author):

Though it is a proof-of-concept study, the feasibility of a real case application should be considered. The whole process/system reported in this work is very complicated. Though the authors believe it is simple, still it is a combination of several processes. The authors should also provide a more detailed analysis and at least compare their methods with many other emerging and existing technologies in wastewater treatment. As the reviewer commented, if it is economically feasible, it should have been applied already. The concept is not new, as it has been proposed or at least discussed in the past. The references provided by the authors also confirmed that the concept is not new. The new information is that the authors tested the concept in the lab, but the results obtained open many uncertainties. The novelty and the quality of this work don't reach the level of being published in this journal. The reviewer would suggest the authors submit it to more technical journals with a more specific readership.

Response: Reviewer #3 maintained that the research work is not new, but failed to bring to our attention any prior work of a similar nature. The reviewer referred to references we cited (without highlight which one) as evidence, but, to the best of our knowledge, none of the papers we cited (for various purposes) reported anything that is related to the concept we proposed and demonstrated in this work. The reviewer further inferred that the concept cannot be economically feasible, completely disregarding our economic analysis without pointing out any flaws in our analysis, based on the argument that 'it should have been applied already' if it were economically feasible. The reviewer requested that the authors "should provide a more detailed analysis and at least compare their methods with many other emerging and existing technologies in wastewater treatment". The fact is that we have done comprehensive comparison with existing technologies via parallel experiments, which was fully recognised by both Reviewer#1 and Reviewer#2.

In view of these comments, we have thoroughly reviewed the paper again, and made several new amendments to further reduce confusions.

Line 71-73: In this work, we propose and demonstrate an electrochemical method for manufacturing iron salts, a solution that effectively addresses two challenges simultaneously. The proposed method is fundamentally different from the existing method of chemical iron salts production.

Line 340-342: In addition, this electrochemical cell can be integrated into existing wastewater treatment systems, following the AD process.

Line 370-371: The low reactivity of C-FeCO₃ limits its applications in urban water management.